



# Changes in soil properties in a low-quality broadleaf mixed forest after cutting strip reforms in a 9-year period in Northeastern China

Huiwen Guan, Xibin Dong , Tian Zhang, Yuan Meng, Jiafu Ruan, Zhiyong Wang

Key Laboratory of Sustainable Forest Management and Environmental Microorganism Engineering of
Heilongjiang Province, Northeast Forestry University, Harbin, 150040, China

*Correspondence to*: Xibin Dong (xibindong@nefu.edu.cn)

**Abstract.** Strip reforms with widths of 6 m, 10 m, 14 m and 18 m were carried out in a low-quality broadleaf mixed forest in Greater Khingan Mountains. The influence of time on soil properties, including physical and chemical properties, were analysed on the basis of data of the soil components obtained from nine consecutive years (from 2010 to 2018). In the
meantime, a principal component analysis was used to determine the weight of each soil indicator, and the fuzzy comprehensive index method was applied to provide further insight into the variation of soil quality. We found that most soil physical properties were damaged by cutting within 3 years and can be restored after 6 years. Over the 9 years, soil physical properties displayed some differences across cutting strip widths, while chemical properties did not display any differences. In terms of chemical properties, they need more time to recover. In view of the current research years, the soil quality could
not be restored in the 18-m harvesting zone within nine years. The cutting width of 10 m is more obvious than that of other transformation widths, so 10 m is the best width for cutting strips for the forest. The study provides reference for the production management of broadleaf mixed forests in the region and other similar areas. A larger width of the cutting strip should be forbidden for this type of forest here. Moreover, for forest soil conditions, we need to continue long-term observations.

## 1 Introduction

In regard to maintaining the productivity and sustainability of forests, soil is a vital factor. For one thing, soil provides the moisture and nutrients to tree growth and supports trees physically. For another, the litter generated by the growing trees can return a great number of nutrients back to soil, through microbial decomposition. There are many ways to intervene in forests, including logging and planting, which usually affect soil nutrients. Cutting timber can heavily impact soil
compaction, temperature and diurnal fluctuation, causing changes in soil (Camenzind et al., 2018;DeLuca and Aplet, 2008). Excessive cutting may result in some consequences that are too ghastly to contemplate, such as forest degradation or soil erosion. In contrast, appropriate harvesting promotes soil nutrients in the long run through complex microbial decomposition (Jamroz et al., 2014;Ma et al., 2013;Zhou et al., 2015).

Many scholars have done lots of related work to reveal the relationship between timber and forest soil. (Guan et al.,
2018;Gao et al., 2013). Recently, many researchers have attached great importance to understanding the impacts(Arevalo-Gardini et al., 2015;Pang et al., 2011). Some studies have shown that harvesting not only destroys the physical properties of the soil, but also affects the chemical properties, especially the soil water holding capacity and soil porosity. (Caldato et al., 2016;Parfitt et al., 2014;Yang et al., 2016); the soil bulk density was increasing and soil was being eroded. (Gerke and





Hierold, 2012;Hieke and Schmidt, 2013;Zhou et al., 2010); and organic matter, N (nitrogen), P (phosphorus), K (potassium), and other minerals were also reduced after cutting. (Ikurekong and Akpabio, 2005;Ong et al., 2012;Ozcan and Gokbulak, 2015).

Some others have found the change of forest stand structure after lumbering(De Nicola et al., 2017;Oyen and Nilsen, 2004;Zhirin and Knyazeva, 2012). Forest ecosystems can be disturbed by cutting heavily. Many trees have been taken away, and the composition of tree species also becomes different. Some have also tried to determine the effects of timber harvesting on biodiversity(Barna and Bosela, 2015;Dechene and Buddle, 2009;Okonogi and Fukuda, 2017). These studies have explored that high-intensity interference may adversely affect biodiversity, while low-intensity interference may benefit biodiversity for a long time. However, most of these studies focused on the short- or medium-term effects of plantation and timber harvesting, in part because of the lack of long-term data. In addition, most of them analyze part of the soil properties rather than the total properties. Therefore, it is necessary to reveal the effects of wood harvesting on the overall properties of the soil in mixed forests over a longer period of time. The study aims to check the impact of cutting strip reforms in the broadleaf mixed forest located in the Daxing'anling mountain range, Northeast China. We try to reveal the change of soil quality in 3, 6 and 9 years after cutting on both physical and chemical properties. Focusing on the impacts of cutting strip reforms, we would like to enrich the existing literature basing on the impacts of cutting strip reforms, which is centred on cutting intensity in forest plantations over a certain time.

In addition, we use fuzzy mathematics and multivariate statistical analyses (such as PCA) to calculate the comprehensive index of soil quality and evaluate its soil quality. Since cutting also changes several soil properties simultaneously while these properties usually interact mutually, it is crucial to collectively reflect the aggregation effect. At last, we explored the effects of various width of strips in broad-leaved mixed forests ranging from 0 m to 18 m with clear cutting. Consequently, our results and conclusion can help determine the optimal width of the cutting strip for the forests in the region. At the same time, our finding can also benefit other regions with similar forests given the geographic spread.

## 2 Study area and methods

### 2.1 Study area

The study area was set on the Yuejin Forest Centre, Jiagedaqi Forestry Bureau, Heilongjiang Province, Northeastern China (124°23'48″-124°24'35″E, 50°34'9″-50°34'32″N). The research plots were established in compartment 174. The elevation of the site ranges from 429 to 521 m with a slope of 6-10°. This area has a cold temperate land monsoon climate. The mean annual temperature is -1.3℃, and the annual precipitation is 494.8 mm. The frost-free period is approximately 85-130 days. According to United States Department of Agriculture (USDA) soil taxonomy, the soil on the study area is classified as brown earth. The thickness of soil is 15-30 cm.

The main tree species are *Quercus mongolica* Fisch. ex Ledeb., *Populus davidiana* Dode, *Betula dahurica* Pall., and *Betula platyphylla* Suk. Shrub species on the site are dominated by *Rhododendronea*, covering 12% of the area. Underground herbaceous and liana species are dominated by *Cyperus microiria* and *Pyrola dahurica*, respectively, covering 27% of the area.

### 2.2 Plot establishment and measurements

In March 2009, cutting strips were established in the low-quality broadleaf mixed forest with the widths of 6 m (S1), 10 m (S2), 14 m (S3), and 18 m (S4) (Figure 1). The length of the transformation zone was 300 m. When cutting the timbers, mature trees were cut down while the coniferous seedlings and rare tree species were preserved. Every cutting strip was divided into three parts (A, B, C) with lengths of 100 m, cultivating *Larix gmelinii* (Rupr.) Kuzen., *Pinus sylvestris* L.var.



*mongolica* Litv., *Pinus koraiensis* Sieb. et Zucc., respectively. In the figure, the bank parts mean harvesting area. The shadow part of the image is the reserved band, and the bandwidth of the reserved band is the same as the bandwidth of the corresponding transformation band at 6 m, 10 m, 14 m and 18 m. The control plot was set up in the same forest without cutting.

The cutting operation consisted of chainsaw cutting, on-site delimbing and bucking, skidding by human shoulder, and collecting and utilizing branches >5 cm in diameter. This logging method is a common practice in the region, and the width is the most important difference between strips. In August 2012 (3 years after the cutting), August 2015 (6 years after cutting) and August 2018 (9 years after the cutting), we measured the characters of the forest, such as the height and DBH. And soil was gathered in different strips and did the experiment in laboratory. Because of the limitations of technical means and

experimental conditions 10 years ago, we only set up a test area in Greater Khingan Mountains. This may lack the necessary sample repetition for the overall situation of the broadleaf mixed forest in the Greater Khingan Mountains. However, this experiment can reflect the soil changes of the current plot to a certain extent, and provide some reference for future research. In order to meet the statistical needs, in other words, to make the sampling point distribution as uniform as possible, we divided each treatment into three parts. In fact, these three parts have been replanted with three species, but this is not

meaningful for this experiment. We do not analyze the differences in soil between different replanted trees (the difference is small). We simply compiled soil samples from the three areas for the composite. The average and standard deviation of the soil content of each treatment plot were obtained by analyzing the results of multiple soil samples according to the random sampling of the soil. Unfortunately, we missed pre-cutting data, so we can't compare this with the data of 3, 6 and 9 years after cutting. If there is this data, the experimental setup will be more perfect. In order to compensate for the defects, we

looked for a non-cutting plot similar to the treatment site conditions and stand composition in the vicinity of the control plot as pre-cutting.

**2.3 Soil sample measurement**

Since the effect of cutting on soil is mostly on surface soil, only the surface soil layers between 0 and 10 cm and between 10 and 20 cm were gathered as sample. The sampling was implemented based on the national standard for gathering and

handling soil samples in forest (Zheng et al., 2008). A soil sample was taken from each of the A, B, and C sections in each plot. Every strip carried 5 soil samples. In order to test physical properties, the soil samples were held in their initial shapes by placing them into aluminium boxes to prevent them from being squeezed and becoming deformed. In order to analyze chemical properties of soil samples, they were put inside plastic bags, sealed and labelled. The three soil samples respectively from the A, B, and C sections in each plot were evenly mixed and air-dried; then, the mixed soil samples were

used for experiment in lab. Therefore, the soil testing results got finally are the representative for the average value from the A, B, and C sections in each strip relatively.

The soil physical properties analysed here included soil bulk density, soil maximum water-holding capacity, soil capillary water-holding capacity, soil non-capillary porosity, soil capillary porosity and soil total porosity because they are vital indicators of soil structure. Meanwhile, indicators of soil chemical properties commonly, such as organic matter, nitrogen

(N), phosphorus (P), and potassium (K), were also considered in this study.

According to the national standard/protocol, analyses of soil physical and chemical properties were done.(Zhang et al., 1984). The water holding capacity was analysed with the cutting ring method (LY/T1215-1999) (Forestry); organic matter was quantified with the potassium dichromate oxidation-external heating method (LY/T 1237-1999) (Forestry); total nitrogen was assessed via the perchloric acid-sulfuric acid digestion diffusion absorption method (LY/T 1228-1999) (Forestry);

water-soluble nitrogen was extracted with the alkaline hydrolysis-diffusion absorption method (LY/T 1229-1999) (Forestry); total phosphorus was estimated with the perchloric acid-sulfuric acid-soluble Mo-Sb colorimetry method (LY/T 1232-1999) (Forestry); rapidly available phosphorus was gauged with the hydrochloric acid-ammonium fluoride extraction method (LY/T 1233-1999) (Forestry); total potassium was measured with the sodium hydroxide alkali fusion-flame photometry



method (LY/T 1234-1999) (Forestry); and rapidly available potassium was tested with the ammonium acetate extraction-flame photometry method (LY/T 1236-1999) (Forestry). The average value of these test results for a given cutting strip individually was to represent its effect on soil physical and chemical properties.

**2.4 Data analysis**

5   With the data derived from laboratory experiment and pre-processing, the percentage change in soil physical and chemical properties was calculated under different cutting strip widths relative to non-cutting. That is, we computed the R-score as follows, where $S_{ij}$ is the mean value of soil property $i$ at the study area with different strip width $j$, and $S_{i0}$ is the mean value of soil property $i$ in the non-cutting plots. The percentage change (R-score) reflects the change in soil properties at a specific width when compared to non-cutting some (3, 6 and 9) years after the reforms by following Eq. (1)

$$R = \frac{(S_{ij} - S_{i0})}{S_{i0}} \times 100\% ,\tag{1}$$

Actually, the aggregate effect of the cutting strip width was the results particularly interested in, which called for a multivariate analysis. Nevertheless, possible correlations among different variables in this model brought statistical complications (Melquiades et al., 2013). To overcome this challenge, we adopted fuzzy mathematics and a principal component analysis.

15   Because of different attributes and dimensions of diverse soil quality indicators, they must be processed before soil quality can be comprehensively evaluated. Data standardization is a statistical method to compare different dimensions and different types of set indicators (Fan et al., 2015). The method follows three principles: the relative difference of data within the same index remains unchanged, the relative difference between different indices remains unchanged, and the maximum value after standardization is equal. In this study, first, the soil physical and chemical properties were standardized and transformed into 20   dimensionless values between 0 and 1, to normalize the dimensions of the indicators. In data standardization, data are divided into 2 types: positive and negative effects. In this study, except for soil bulk density, the other indicators are positive effects. The positive and negative effects are calculated by Eq. (2) and Eq. (3) respectively. That is, we computed $F(X_i)$, which is the membership value of soil property $i$, reflecting the evaluation as follows, where $X_{i\max}$ is the maximum value of soil property $i$; $X_{ij}$ is the average value of the measured sample of soil property $i$; and $X_{i\min}$ is the minimum value of soil 25   property $i$.

$$F(X_i) = (X_{i\max} - X_{ij}) / (X_{i\max} - X_{i\min}) ,\tag{2}$$

$$F(X_i) = (X_{ij} - X_{i\min}) / (X_{i\max} - X_{i\min}) ,\tag{3}$$

Because the importance of each factor is different, namely, the degree of impact on soil quality is diverse, it needs to be given distinct weights. In this study, statistical software is used to analyse the standardized data of 13 indicators using a 30   principal component analysis, and the contribution rate and cumulative contribution rate of each factor are calculated. The load matrix is obtained by common factor rotation, and the common factor variance of the soil quality index is calculated to show its contribution to the overall variation of soil quality. The proportion of the common factor variance of each index to the total common factor variance is taken as the weight of each index.



Based on the evaluation factors of membership degree and weight determination, using the weighted method and addition rule in fuzzy mathematics, we use Eq. (4) to calculate the soil quality of different cutting strip widths in these years. *F* is the comprehensive index of soil quality, and $W_i$ is the weight of each soil factor, which reflects the importance of each evaluation index.

$$F = \sum W_i \times F(X_i) \ ,$$ (4)

## 3 Results

### 3.1 Impacts on soil physical properties individually

As shown in Table 1, it was followed that the measurements of soil physical properties at the study area under different cutting strip widths in 3, 6 and 9 years after cutting. All of these indicators showed a certain variation trend, indicating that
soil physical properties could be at least influenced over time. However, all these changes were not dramatic. Soil bulk density also showed a further decline from 6 to 9 years after the cutting in most cutting strips. In different years, there were differences in the correlation between the changes of indices and the width of cutting strips. Three years after cutting, soil bulk density decreased and then increased; however, other properties increased and then decreased with an increase in cutting strip width. After 6 and 9 years after cutting, the variation trends were similar, but there was diversity in the turning
points.

According to the R-scores (Table 2), the soil bulk density of the 18 m cutting strip was significantly higher than that of the control plot. At the same time, most other soil indices in each cutting zone were lower than those in the control plot, indicating that soil properties were disturbed after 3 years. The soil maximum water-holding capacity, soil capillary porosity and total soil porosity in the 6 m and 10 m cutting strips were larger than those in the uncultivated plots, which indicated that
the physical properties of the reformed zones were developing in a good direction in 6 years. The soil bulk density of the 18-m cutting strip was still significantly higher than that of the control plot, and other index values were lower than that of the control plot after 9 years. It may be that the width of 18 m was too wide to restore soil properties after 9 years. Several indices of other width zones are superior to the control plots, so it is difficult to judge the overall effect of the reconstruction.

Recovery or deterioration rates of soil physical properties also displayed some differences across cutting strip widths. For
example, the soil capillary water-holding capacity, soil capillary porosity and soil total porosity were restored faster under these reforms than under non-cutting between 3 and 9 years after cutting. However, between 3 and 9 years after cutting, soil bulk density decreased faster only under the widths of 10 m and 14 m than under non-cutting, whereas the soil maximum water-holding capacity increased only under 6-m and 14-m cutting strips than under non-cutting. Similarly, the growth of soil non-capillary porosity was significantly higher under 6-m, 10-m, and 14-m cutting strips than that of control plots (non-
cutting).

Table 1. Soil physical properties in 3, 6 and 9 years after cutting

| Cutting Strip Width | Soil Bulk Density (g·cm⁻³) | Soil Maximum Water-holding Capacity (%) | Soil Capillary Water-holding Capacity (%) | Soil Non-capillary Porosity (%) | Soil Capillary Porosity (%) | Soil Total Porosity (%) |
|---|---|---|---|---|---|---|





| | | | 3 years after cutting | | | |
|---|---|---|---|---|---|---|
| 6 m | 0.63±0.08 | 92.63±16.65 | 81.25±13.47 | 7.41±1.97 | 52.9±7.90 | 60.31±10.87 |
| 10 m | 0.62±0.11 | 96.56±16.76 | 84.10±10.24 | 7.03±2.07 | 55.45±7.50 | 62.48±9.65 |
| 14 m | 0.66±0.11 | 89.51±14.36 | 79.99±12.57 | 8.22±2.62 | 52.03±6.78 | 60.25±9.29 |
| 18 m | 0.72±0.19 | 75.36±16.31 | 58.06±12.56 | 13.13±2.13 | 44.08±7.87 | 57.21±7.02 |
| Non-cutting | 0.66±0.09 | 101.47±16.12 | 98.88±10.74 | 13.01±1.80 | 53.48±8.80 | 66.49±8.62 |
| | | | 6 years after cutting | | | |
| 6 m | 0.57±0.09 | 109.26±17.24 | 85.25±13.94 | 20.23±2.64 | 46.38±9.66 | 66.61±11.25 |
| 10 m | 0.60±0.12 | 106.58±14.98 | 89.23±11.67 | 16.42±2.38 | 51.37±6.28 | 67.79±9.28 |
| 14 m | 0.65±0.12 | 81.23±16.43 | 83.12±9.90 | 9.12±1.39 | 54.81±9.24 | 63.93±10.96 |
| 18 m | 0.63±0.16 | 82.14±19.83 | 78.37±11.3 | 10.58±1.79 | 50.11±7.74 | 60.69±11.37 |
| Non-cutting | 0.64±0.11 | 105.47±12.86 | 89.64±11.21 | 15.29±1.80 | 49.89±7.95 | 65.18±8.88 |
| | | | 9 years after cutting | | | |
| 6 m | 0.61±0.11 | 117.43±17.59 | 85.21±15.67 | 13.97±1.47 | 52.47±7.02 | 66.44±7.93 |
| 10 m | 0.59±0.12 | 106.46±13.27 | 88.47±14.82 | 11.29±2.16 | 54.13±7.23 | 65.42±11.02 |
| 14 m | 0.62±0.11 | 85.14±13.30 | 81.26±12.82 | 9.87±1.79 | 53.14±8.93 | 63.01±8.38 |
| 18 m | 0.69±0.14 | 79.68±16.01 | 80.12±13.06 | 10.13±1.54 | 48.35±8.60 | 58.48±10.79 |
| Non-cutting | 0.63±0.11 | 108.47±19.55 | 91.02±13.45 | 13.51±2.07 | 51.23±8.40 | 64.74±8.04 |

Note: The number in the table is "average ±standard deviation". Different letters in each column indicate significant differences ($P<0.05$).

Table 2. Percentage changes in soil physical properties due to different cutting strip widths relative to non-cutting

| Soil Property | Cutting Strip Width (3 Years after Cutting) | | | | Cutting Strip Width (6 Years after Cutting) | | | | Cutting Strip Width (9 Years after Cutting) | | | |
|---|---|---|---|---|---|---|---|---|---|---|---|---|
| | 6 m | 10 m | 14 m | 18 m | 6 m | 10 m | 14 m | 18 m | 6 m | 10 m | 14 m | 18 m |
| Soil Bulk Density | -4.55 | -6.06 | 0.00 | 9.09 | -10.94 | -6.25 | 1.56 | -1.56 | -3.17 | -6.35 | -1.59 | 9.52 |
| Soil Maximum Water-holding Capacity | -8.71 | -4.84 | -11.79 | -25.73 | 3.59 | 1.05 | -22.98 | -22.12 | 8.26 | -1.85 | -21.51 | -26.54 |
| Soil Capillary Water-holding Capacity | -17.83 | -14.95 | -19.10 | -41.28 | -4.90 | -0.46 | -7.27 | -12.57 | 6.38 | -2.80 | -10.72 | -11.98 |
| Soil Non-capillary Porosity | -43.04 | -45.96 | -36.82 | 0.92 | 32.31 | 7.39 | -40.35 | -30.80 | 3.40 | 16.43 | 26.94 | -25.02 |
| Soil Capillary Porosity | -1.08 | 3.68 | -2.71 | -17.58 | -7.04 | 2.97 | 9.86 | 0.44 | 2.42 | 5.66 | 3.73 | -5.62 |
| Soil Total Porosity | -9.29 | -6.03 | -9.38 | -13.96 | 2.19 | 4.00 | -1.92 | -6.89 | 2.63 | 1.05 | -2.67 | -9.67 |

Table 3. Percentage change in soil physical properties

| Soil Property | Between 3 and 6 years after cutting | | | | | Between 6 and 9 years after cutting | | | | | Between 3 and 9 years after cutting | | | | |
|---|---|---|---|---|---|---|---|---|---|---|---|---|---|---|---|
| | 6 m | 10 m | 14 m | 18 m | Non-cutting | 6 m | 10 m | 14 m | 18 m | Non-cutting | 6 m | 10 m | 14 m | 18 m | Non-cutting |
| Soil Bulk Density | -9.52 | -3.23 | -1.52 | 12.50 | -3.03 | 7.02 | -1.67 | 4.62 | 9.52 | -1.56 | -3.17 | -4.84 | -6.06 | -4.17 | -4.55 |
| Soil Maximum Water-holding Capacity | 17.95 | 10.38 | -9.25 | 9.00 | 3.94 | 7.48 | -0.11 | 4.81 | -2.99 | 2.84 | 26.77 | 10.25 | -4.88 | 5.73 | 6.90 |
| Soil Capillary Water-holding Capacity | 4.92 | 6.10 | 3.91 | 34.98 | -9.34 | -0.05 | -0.85 | 2.24 | -2.23 | 1.54 | 4.87 | 5.20 | 1.59 | 38.00 | -7.95 |



| | | | | | | | | | | | | | | | |
|---|---|---|---|---|---|---|---|---|---|---|---|---|---|---|---|
| Soil Non-capillary Porosity | 173.01 | 133.57 | 10.95 | 19.42 | -17.52 | -30.94 | -31.24 | 8.22 | -4.25 | -11.64 | 88.53 | 60.60 | 20.07 | -22.85 | 3.84 |
| Soil Capillary Porosity | -12.33 | -7.36 | 5.34 | 13.68 | -6.71 | 13.13 | 5.37 | -3.05 | -3.51 | 2.69 | -0.81 | -2.38 | 2.13 | 9.69 | -4.21 |
| Soil Total Porosity | 10.45 | 8.50 | 6.11 | 6.08 | -1.97 | -0.26 | -3.50 | -1.44 | -3.64 | -0.68 | 10.16 | 4.71 | 4.58 | 2.22 | -2.63 |

### 3.2 Impacts on soil chemical properties individually

As shown in Table 4, it was followed that measurements of soil chemical properties at the study area under different cutting strip widths in 3, 6 and 9 years after the cutting. Overall, the properties' values increased and then decreased with an increase in cutting strip width. In different years after cutting, the peak value of each index didn't appear in the same strip.

According to the R-scores (Table 5), all the chemical properties of cutting strips were significantly higher than those of the uncultivated plots after 3 years, which may be due to the accumulation of a large number of soil chemical elements. After 6 years, the contents of organic matter, rapidly available phosphorus and rapidly available potassium in 6 m and 18 m cutting strips were lower than those in control plots. At the same time, the total nitrogen and total potassium content of the 18 m cutting strip were also slightly lower than that of control plots. After 9 years, total phosphorus, rapidly available phosphorus

and rapidly available potassium in the 6-m cutting strip were lower than those in control plot. Meanwhile, total nitrogen, rapidly available phosphorus and total potassium in the 18-m cutting strip were also lower than those in control plot.

   Recovery or deterioration rates of soil chemical properties did not display significant differences across cutting strip width. During the nine years after cutting, the change of soil nutrients was not significant, and the change of each cutting strip was basically the same as that of the unharvested sample base. Compared with the third year after cutting, the index value of the

cutting strip decreased mostly after 9 years. This decrease may be due to the relatively long process of changing soil chemical indicators, which did not produce significant results in nine years, and the effect of litter accumulation after harvesting was weakened.

Table 4. Soil chemical properties in 3, 6 and 9 years after cutting

| Cutting Strip Width | Organic Matter (g·kg⁻¹) | Total Nitrogen (g·kg⁻¹) | Water-soluble Nitrogen (mg·kg⁻¹) | Total Phosphorus (g·kg⁻¹) | Rapidly Available Phosphorus (mg·kg⁻¹) | Total Potassium (g·kg⁻¹) | Rapidly Available Potassium (mg·kg⁻¹) |
|---|---|---|---|---|---|---|---|
| *3 years after cutting* | | | | | | | |
| 6 m | 21.25±2.65 | 9.25±2.00 | 519.63±67.02 | 2.20±0.21 | 14.86±2.09 | 9.24±1.45 | 54.32±7.50 |
| 10 m | 22.90±2.77 | 9.45±2.86 | 545.01±53.27 | 2.23±0.24 | 16.13±2.37 | 9.36±2.29 | 56.05±7.19 |
| 14 m | 22.19±3.00 | 9.21±3.39 | 557.92±62.90 | 2.41±0.30 | 16.24±2.60 | 10.21±1.89 | 58.48±8.14 |
| 18 m | 24.28±3.28 | 9.93±1.7 | 530.28±59.48 | 2.38±0.24 | 15.86±2.34 | 9.35±1.37 | 58.13±7.28 |
| Non-cutting | 20.95±1.69 | 8.58±1.24 | 469.81±60.38 | 2.13±0.22 | 13.9±1.23 | 9.11±2.12 | 55.87±6.91 |
| *6 years after cutting* | | | | | | | |
| 6 m | 21.32±2.34 | 8.82±2.54 | 481.39±49.53 | 2.08±0.38 | 13.42±2.25 | 9.02±1.66 | 53.11±5.28 |
| 10 m | 21.90±2.24 | 9.26±1.53 | 512.36±49.65 | 2.11±0.24 | 14.97±2.07 | 9.14±1.67 | 57.12±6.82 |
| 14 m | 22.59±2.83 | 8.91±2.83 | 500.31±53.84 | 2.24±0.21 | 14.82±1.89 | 9.57±2.16 | 55.51±10.21 |
| 18 m | 21.53±1.56 | 8.74±2.09 | 492.13±53.83 | 2.10±0.40 | 13.31±1.39 | 8.81±2.09 | 52.57±8.89 |
| Non-cutting | 21.61±2.24 | 8.76±2.42 | 468.24±66.89 | 2.07±0.33 | 13.81±1.70 | 8.86±2.50 | 54.12±9.53 |
| *9 years after cutting* | | | | | | | |
| 6 m | 21.85±1.93 | 8.79±2.28 | 485.69±65.71 | 2.09±0.30 | 13.85±2.26 | 9.01±1.80 | 53.17±6.79 |
| 10 m | 22.15±2.28 | 9.38±1.8 | 521.31±75.66 | 2.16±0.31 | 15.23±2.69 | 9.14±2.19 | 58.14±6.54 |
| 14 m | 21.57±2.85 | 8.98±2.17 | 507.46±64.99 | 2.31±0.31 | 14.98±1.87 | 9.82±2.29 | 55.74±7.23 |
| 18 m | 21.94±1.69 | 8.71±1.94 | 497.52±63.68 | 2.14±0.24 | 13.72±2.01 | 8.85±2.40 | 52.48±10.55 |
| Non-cutting | 21.38±2.27 | 8.79±2.49 | 471.21±63.05 | 2.12±0.27 | 13.94±2.4 | 8.92±1.78 | 55.46±6.03 |





Note: The number in the table is "average ± standard deviation". Different letters in each column indicate significant differences ($P<0.05$).

Table 5. Percentage changes in soil chemical properties due to different cutting strip widths relative to non-cutting

| Soil Property | Cutting Strip Width (3 Years after Cutting) | | | | Cutting Strip Width (6 Years after Cutting) | | | | Cutting Strip Width (9 Years after Cutting) | | | |
|---|---|---|---|---|---|---|---|---|---|---|---|---|
| | 6 m | 10 m | 14 m | 18 m | 6 m | 10 m | 14 m | 18 m | 6 m | 10 m | 14 m | 18 m |
| Organic Matter | 1.43 | 9.31 | 5.92 | 15.89 | -1.34 | 1.34 | 4.53 | -0.37 | 2.20 | 3.60 | 0.89 | 2.62 |
| Total Nitrogen | 7.81 | 10.14 | 7.34 | 15.73 | 0.68 | 5.71 | 1.71 | -0.23 | 0.00 | 6.71 | 2.16 | -0.91 |
| Water-soluble Nitrogen | 10.60 | 16.01 | 18.75 | 12.87 | 2.81 | 9.42 | 6.85 | 5.10 | 3.07 | 10.63 | 7.69 | 5.58 |
| Total Phosphorus | 3.29 | 4.69 | 13.15 | 11.74 | 0.48 | 1.93 | 8.21 | 1.45 | -1.42 | 1.89 | 8.96 | 0.94 |
| Rapidly Available Phosphorus | 6.91 | 16.04 | 16.83 | 14.10 | -2.82 | 8.40 | 7.31 | -3.62 | 0.65 | 9.25 | 7.46 | -1.58 |
| Total Potassium | 1.43 | 2.74 | 12.07 | 2.63 | 1.81 | 3.16 | 8.01 | -0.56 | 1.01 | 2.47 | 10.09 | -0.78 |
| Rapidly Available Potassium | -2.77 | 0.32 | 4.67 | 4.05 | -1.87 | 5.54 | 2.57 | -2.86 | -4.13 | 4.83 | 0.50 | -5.37 |

Table 6. Percentage change in soil chemical properties

| Soil Property | Between 3 and 6 years after cutting | | | | | Between 6 and 9 years after cutting | | | | | Between 3 and 9 years after cutting | | | | |
|---|---|---|---|---|---|---|---|---|---|---|---|---|---|---|---|
| | 6 m | 10 m | 14 m | 18 m | Non-cutting | 6 m | 10 m | 14 m | 18 m | Non-cutting | 6 m | 10 m | 14 m | 18 m | Non-cutting |
| Organic Matter | 0.00 | -0.04 | 0.02 | -0.11 | 0.03 | 0.02 | 0.01 | -0.05 | 0.02 | -0.01 | 0.03 | -0.03 | -0.03 | -0.10 | 0.02 |
| Total Nitrogen | -0.05 | -0.02 | -0.03 | -0.12 | 0.02 | 0.00 | 0.01 | 0.01 | 0.00 | 0.00 | -0.05 | -0.01 | -0.02 | -0.12 | 0.02 |
| Water-soluble Nitrogen | -0.07 | -0.06 | -0.10 | -0.07 | 0.00 | 0.01 | 0.02 | 0.01 | 0.01 | 0.01 | -0.07 | -0.04 | -0.09 | -0.06 | 0.00 |
| Total Phosphorus | -0.05 | -0.05 | -0.07 | -0.12 | -0.03 | 0.00 | 0.02 | 0.03 | 0.02 | 0.02 | -0.05 | -0.03 | -0.04 | -0.10 | 0.00 |
| Rapidly Available Phosphorus | -0.10 | -0.07 | -0.09 | -0.16 | -0.01 | 0.03 | 0.02 | 0.01 | 0.03 | 0.01 | -0.07 | -0.06 | -0.08 | -0.13 | 0.00 |
| Total Potassium | -0.02 | -0.02 | -0.06 | -0.06 | -0.03 | 0.00 | 0.00 | 0.03 | 0.00 | 0.01 | -0.02 | -0.02 | -0.04 | -0.05 | -0.02 |
| Rapidly Available Potassium | -0.02 | 0.02 | -0.05 | -0.10 | -0.03 | 0.00 | 0.02 | 0.00 | 0.00 | 0.02 | -0.02 | -0.04 | -0.05 | -0.10 | -0.01 |

## 3.3 Impacts on overall soil properties

### 3.3.1 Determining the Weights of Indices

Eq. (2) and Eq. (3) were used to standardize the data of 13 soil quality indicators, and then a principal component analysis (PCA) was used to calculate the contribution rate and cumulative contribution rate of each factor. The load matrix was obtained by the common factor rotation, the common factor variance of soil quality index was calculated, and the weight was calculated. The results of the principal component analysis and weight accounting of 13 soil quality indicators are shown in Table 7.

According to Table 7, we can see that the eigenvalue of the first principal component is 6.72, which accounts for 51.71% of the total variance. The cumulative contribution rate of the three principal component factors extracted was 85.27%, which almost contained all the information of the original data and was in accordance with the condition that the cumulative contribution rate of principal component analysis was more than 80%.

Table 7. Rotated principal component matrix, communality and weight of each indicator

| Index | Principal Component | $\sigma^2$ of common | Weight |
|---|---|---|---|





|  | 1 | 2 | 3 | factor | |
|---|---|---|---|---|---|
| Soil Bulk Density | -0.5668 | 0.5585 | 0.2208 | 0.6819 | 0.0615 |
| Soil Maximum Water-holding Capacity | -0.6869 | 0.4754 | 0.4092 | 0.8652 | 0.0781 |
| Soil Capillary Water-holding Capacity | -0.7444 | 0.5511 | -0.1933 | 0.8952 | 0.0808 |
| Soil Non-capillary Porosity | -0.5962 | -0.1960 | 0.7170 | 0.9080 | 0.0819 |
| Soil Capillary Porosity | -0.0617 | 0.7962 | -0.5147 | 0.9027 | 0.0814 |
| Soil Total Porosity | -0.7277 | 0.5516 | 0.3047 | 0.9266 | 0.0836 |
| Organic Matter | 0.7849 | -0.1836 | 0.3795 | 0.7938 | 0.0716 |
| Total Nitrogen | 0.7950 | 0.1269 | 0.4835 | 0.8818 | 0.0796 |
| Water-soluble Nitrogen | 0.8592 | 0.3197 | 0.0383 | 0.8419 | 0.0759 |
| Total Phosphorus | 0.9322 | 0.1090 | -0.0744 | 0.8864 | 0.0800 |
| Rapidly Available Phosphorus | 0.8615 | 0.4643 | 0.1417 | 0.9778 | 0.0882 |
| Total Potassium | 0.6960 | 0.4625 | -0.1595 | 0.7238 | 0.0653 |
| Rapidly Available Potassium | 0.6357 | 0.5221 | 0.3507 | 0.7997 | 0.0721 |
| Eigenvalue | 6.72 | 2.68 | 1.68 | | |
| Proportion (%) | 51.71 | 20.65 | 12.91 | | |
| Cumulative proportion (%) | 51.71 | 72.36 | 85.27 | | |

### 3.3.2 Soil Quality Index

On the basis of determining the subordinate degree and weight of evaluation index factors, the comprehensive index of soil quality in different years and different widths of cutting strips was calculated by using the weighted synthesis method and the addition and multiplication rule in fuzzy mathematics. The method is shown in Formula 5. The transformation of soil quality with cutting width in different years is shown in Figure 2.

Soil quality has a great relationship with the width of the cutting strip, and the optimum cutting width has changed in different years. In the third year after cutting, the comprehensive index of soil quality showed the cutting width with 14 m (0.6280) > 10 m (0.6043) > 18 m (0.4844) > non-cutting (0.4195) > 6 m (0.4137). Except for the 6-m transformation zone, the soil quality of other transformation plots was better than that of the control plots, which may be due to the increase of soil nutrients caused by the decomposition of harvested residues. In the 6[th] year after harvesting, the comprehensive index of soil quality was 10 m (0.5913) > 14 m (0.4713) > 6 m (0.4071) > non-cutting (0.3689) > 18 m (0.2327), and the soil quality in the 18-m harvesting zone was significantly lower than that in other modified plots and the control plot. It was possibly that the soil nutrient loss caused by the wide harvesting width could not be restored within 6 years. Nine years after cutting, the relationship between the comprehensive index of soil quality and the cutting strip was basically consistent with that after six years of harvesting, which was 10 m (0.6148) > 14 m (0.4965) > 6 m (0.4071) > non-cutting (0.3689) > 18 m (0.2082). This indicated that the soil quality could not be restored in the 18-m harvesting zone within nine years. It may be that the cutting width is too wide for this experimental stand, or it may take longer to restore soil quality. In view of the current research years, the cutting width of 10 m is more obvious than that of other transformation widths.

### 4 Discussion

Our results showed that the width of the cutting strip had a significant impact on overall soil physical and chemical properties. In general, the soil bulk density decreases and then increases, but soil porosity and water holding capacity increase and then decrease as width increases after many years of cutting reform, echoing the results reported in the literature





(Jennings et al., 2012;Lu, 2006;Makineci et al., 2007). Likewise, an increase in the width of the cutting strip, after years of recovery in our study, could cause a recovery but then loss of soil nutrients (N, P, and K), which is parallel to the finding of existing studies (XU and WEI, 2013;Ying et al., 2012). However, soil nutrients were lost compared with the year nearest cutting, meaning that they possibly need more years to recover.

In addition to confirming existing findings, our study shed new light on the aggregate impact of cutting strips on overall soil properties. The results from PCA revealed that the first principal component was exclusively associated with soil nutrients, which explained most variation in the impact of cutting strips, and the second principal component was mostly linked to soil physical properties. Therefore, people are most concerned about the loss of soil nutrients (especially phosphorus and potassium) due to the excessive width of cutting strips in the forest, which has a negative impact on soil physical properties
and has difficulty recovering in a short time. Without nutrient supplementation, if not fertilized, soil nutrient loss will reduce long-term soil productivity and lead to forest degradation.

Moreover, a certain width of cutting strip can promote soil nutrients after a certain year. The recovery of soil properties impacted by most widths of cutting strips is a slow process. It would take longer for overall soil properties to recover as the width rises. This was not only because a wider of cutting strip would cause greater damage to soil properties but also because
the recovery rate of soil properties would slow down sooner with an increase in the cutting strip width. Thus, additional time in our study may not be very helpful in restoring soil properties damaged by an excessive width of cutting. With even more years, soil quality could not be fully restored if the cutting strip is exceeds a normal range.

Given the rising demand for timber and the promotion of stand regeneration, appropriate harvesting from this forest seems necessary. With all the above impacts in mind, if timber harvesting from this forest has to continue to some extent, the width
of cutting strip should be maintained at approximately 10 m. Additionally, it is feasible to supplement nutrients by applying appropriate fertilizers to help regenerate or restore forests in the region.

The effects of cutting band width on soil physical and chemical properties and soil comprehensive quality were studied. In the future, we will explore the effects of cutting band width on stand regeneration and stand structure, not only on soil properties but also on forest productivity and forest resilience. In addition, the application of cutting zones to more tree
species can help us explore the effects of other factors, such as tree species composition and environmental conditions. Finally, it is of great value for the sustainable development of forests to carry out continuous observations in the experimental area and to study the effects of more years of rehabilitation on forests.

## 5 Conclusions

It was examined that the impact of cutting strip width on soil physical and chemical properties in a low-quality broadleaf
mixed forest in northeastern China in 3, 6 and 9 years after cutting reform. We considered four treatments—6 m, 10 m, 14 m, and 18 m widths of cutting strips—with non-cutting as the control. We analysed the impacts of cutting intensity on soil properties both individually and comprehensively. After 9 years, in terms of impacts on individual soil properties, cutting strip reform caused a much greater impact on most soil physical properties than that of non-cutting, while the impact on soil chemical properties was augmented with an increase in cutting strip width. As for aggregate impacts on overall soil physical
and chemical properties, the difference of strip width showed various impact on it.

These findings will make vital  implications for sustainable ecological management to the mixed natural broadleaf forest in the study region and places similarly. First, most soil physical properties are damaged by cutting within 3 years and can be restored after 6 years, while the impact on soil chemical properties increased with an increase in cutting strip width within 3 years. Second, over 9 years, soil physical properties displayed some differences across cutting strip widths, while chemical
properties didn't. Chemical properties needed more time to recover. In view of the current research years, the soil quality





could not be restored in the 18-m harvesting zone within nine years. The cutting width of 10 m is more obvious than that of other transformation widths, so 10 m is the best width of cutting strip for the forest. Hence, a suitable width of the cutting strip can increase soil nutrients after certain years. However, it has a critical value, which means that if we apply wider cutting strip to a forest stand, the soil nutrients cannot be recovered or it takes a long time. Third, given the impacts of cutting strips on both individual and overall soil properties, a large width of the cutting strip in this type of forest in the region should be avoided. For forest soil conditions, we need to continue long-term observations.

*Competing Interest*. The authors declare no conflicts of interest.

*Acknowledgements*. This research was funded by the National Key R&D Program of China (2017YFC0504103) and the Fundamental Research Funds for the Central Universities (2572017AB20).

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

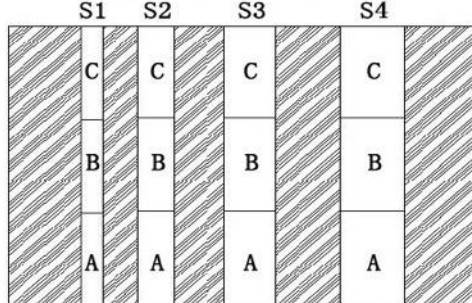

**Figure 1: Strip plots settings**





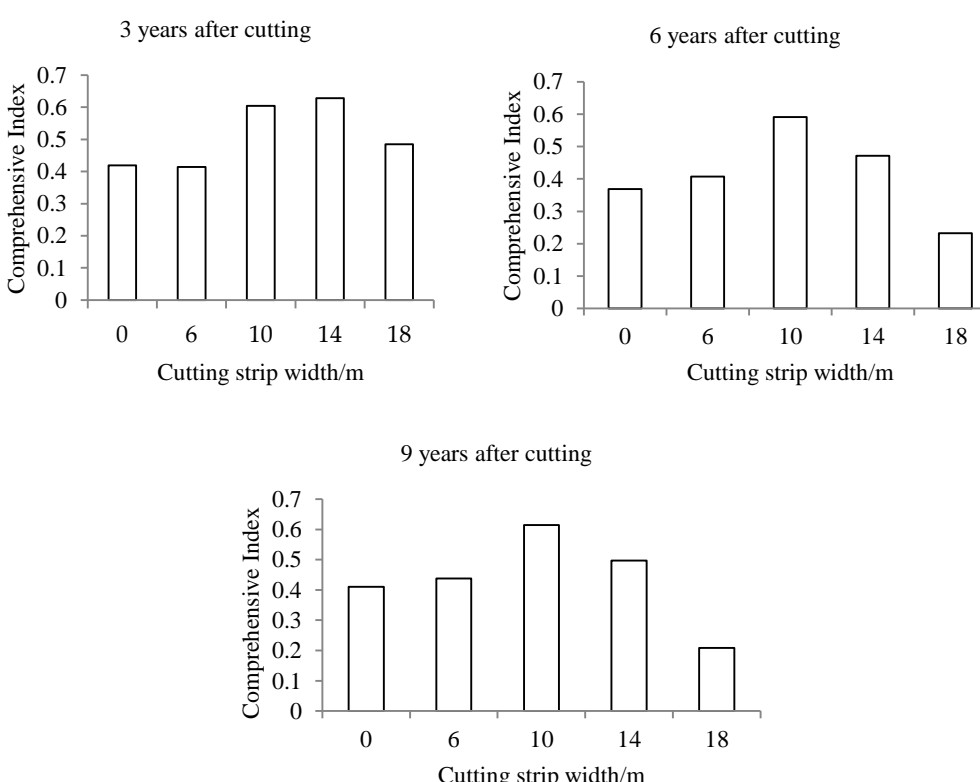

**Figure 2: Comprehensive soil quality index under cutting strips 3, 6 and 9 years after cutting**