# Peer review of "Changes in soil properties in a low-quality broadleaf mixed forest after cutting strip reforms in a 9-year period in Northeastern China"

_SOIL, 2019_

## Referee Comment (RC1) · Anonymous Referee #1 · 5 Jun 2019

The manuscript is not ready to be published in its current state. Some more work is necessary to be done by the authors in order to improve it both, from the scientific and formal/technical point of view. Besides the results from this kind of long-term experiment has a high potential applicability for forest management in low quality environments, the way that the data are analysed and the work presented make it very difficult to understand. Methodological section needs some more detail: the study area and experimental design is not well described: some information in relation to the soils in the area (texture, pH, for example), total area under study (only length is mentioned), number of trees in each plot, sampling date (is the same date all the years?. . .). In relation to the number of samples confusing information is given: 1

sample per plot?, 5 samples per strip? Composed? , three samples mixed???...In addition, it seems that 0-10 cm and 10-20 cm depth samples are taking (page 3, line 23) but this information is not given in tables... Also, if the soil depth is 30 cm (page 2 line 30), why did not do the sampling at this depth??; area occupied for each treatment; you should give some more detailed about the vicinity area selected as pre-cutting conditions (control): for example how far is it? A map indicating where the study area is, etc... should help to understand the experimental design. Some methods, in particular, soil physical properties, should be better described. Please, explain all the variables you are introduced in the analysis... Results are not well explained. Tables are very difficult to interpret. I suggest using figures instead. Also, no statistical analysis is reported. Thus, it is so difficult to interpret the results due the high variability among years and width strips. I would suggest using a general lineal model with repeated measurements ANOVA design and try to separate the effects of years and strip. Also the interactions.. Discussion section is very poor. Some more references and scientific explanations from the results should be included and discussed. There are quite a lot of technical errors through the mnuscript (marked in the annotated file), including the absence of the journals in some articles cited. In the attached file other comments are related.

Please also note the supplement to this comment:
https://www.soil-discuss.net/soil-2019-10/soil-2019-10-RC1-supplement.pdf

**Supplement:**

[revised manuscript text omitted]

---

## Referee Comment (RC2) · Anonymous Referee #2 · 4 Jul 2019

The submitted manuscript "Changes in soil properties in a low-quality broadleaf mixed forest after cutting strip reforms in a 9-year period in Northeastern China" describes the impacts of timber cutting on selected physical and chemical soil properties throughout a nine year period. Such data from long-term experiment are sparse and, therefore, the information given in this paper is valuable for forest management. Nevertheless, the in the current state the manuscript is not ready to be published.

In the Introduction section the objectives need to be outlined more clearly. The authors talk about overall soil properties but they investigated only selected physical and chemical soil parameters. Materials and methods should be explained more in

detail. Number of soil samples and depths of sampling are not mentioned as well as if replications were taken. Soil physical parameters and the methods of analyses should be better described as they are not commonly used in soil physics. Basic soil physical and chemical parameters of the plots should be included. Statistical analyses are performed but the used analyses are not described. Although the results of physical and chemical parameters show no significant differences. The calculated soil quality index is based on 13 indicators which are interacting like total nitrogen, water soluble nitrogen etc. Determining a soil quality index should use indicators which are to interfering each other. The tables are not easy to interpret. I suggest the same layout for all tables or to display the results in figures instead. Results are not well explained and discussion section could be more elaborated. Overall, the manuscript is difficult to understand and needs improvements.

Please also note the supplement to this comment:
https://www.soil-discuss.net/soil-2019-10/soil-2019-10-RC2-supplement.pdf

―――――――――――――――――――――――――――

**Supplement:**

[revised manuscript text omitted]

---

## Referee Comment (RC3) · Anonymous Referee #3 · 4 Jul 2019

In their manuscript the authors describe a study in which they examined the effects of clear-cutting over time on a variety of soil physical and soil chemical factors. In itself this topic of research is relevant and interesting if it is focused on unraveling underlying mechanism / processes responsible for the observed effects. In other words, hypothesis driven research aimed at gaining a fundamental understanding of generic processes. Unfortunately, I must conclude that the current study was not setup in such a manner. Instead, it is a purely descriptive work that presents a lot of measurement data and looks at trends, but makes hardly any effort to understand those trends. This makes the study of limited scientific value.

[Figure]

In addition, the results are presented in a poor and sloppy manner. There are many mistakes in the English, but also in the general layout, font size, etc. that give the feeling that presentation did not get the attention that one can expect be given to a serious submission to a scientific journal. Also the presentation of the data in endless tables rather than informative graphs is below par.

All together, I see no alternative than to recommend rejection of this manuscript for publication in SOIL.

---

## Author Comment (AC1) · 2 Aug 2019

Thanks for your suggestion. I know there still are some problems in this manuscript but I hope you can give me the opportunity to correct it as a beginner. First, I have tried to use a new method called repeated measures ANOVA to analyze the data which will better to understand. Second, I have tried my best to modify the methodological section. I think it will be clearer. But I'm sorry for that some information I can't provide because the raw data doesn't include it and there is no chance to go to field again texting it these days. Third, I tried to use some figures to describe the results and renew the content. What's more, I rewrote the discussion part and read more

references to incite. And I have corrected the wrong of the details in the paper. Have good time.

Please also note the supplement to this comment:
https://www.soil-discuss.net/soil-2019-10/soil-2019-10-AC1-supplement.pdf

**Supplement:**

**Changes in soil properties in a low-quality broadleaf mixed forest after cutting strip reforms in a 9-year period in Northeastern China**

Huiwen Guan, Xibin Dong, Tian Zhang, Yuan Meng, Jiafu Ruan, Zhiyong Wang

Key Laboratory of Sustainable Forest Management and Environmental Microorganism Engineering of Heilongjiang Province,
Northeast Forestry University, Harbin, 150040, China

Correspondence to: Xibin Dong (xibindong@nefu.edu.cn)

**Abstract.** Strip reforms with widths of 6 m, 10 m, 14 m and 18 m were carried out in a low-quality broadleaf mixed forest in Greater Khingan Mountains. The influence of time on soil properties, including physical and chemical properties, were analysed on the basis of data of the soil components obtained from nine consecutive years (from 2010 to 2018). First, use the

- 10 repeated measures ANOVA to distinguish the effects of various width of cutting strip and years. In the meantime, a principal component analysis was used to determine the weight of each soil indicator, and the fuzzy comprehensive index method was applied to provide further insight into the variation of soil quality. We found that most soil physical properties can be affected by strips while only two indicators can be affected by years. And half of the indicators can be affected by the interaction by strip and year. As for soil chemical properties, only two plant available elements (N and P) can be affected
- 15 either strip or year. In addition, no indicator was changed by the interaction. Over the 9 years, soil physical properties displayed more differences than chemical properties across cutting strip widths. However, it's not enough for the properties to recover for 9 years unfortunately. In view of the current research years, the soil quality could not be restored in the 18-m harvesting zone within nine years. The cutting width of 10 m is more obvious than that of other transformation widths, so 10 m is the best width for cutting strips for the forest. The study provides reference for the production management of broadleaf
- 20 mixed forests in the region and other similar areas. A larger width of the cutting strip should be forbidden for this type of forest here. Moreover, for forest soil conditions, we need to continue long-term observations.

**1** Introduction**

In regard to maintaining the productivity and sustainability of forests, soil is a vital factor. For one thing, soil provides the moisture and nutrients to tree growth and supports trees physically. For another, the litter generated by the growing trees can return a great amount of nutrients back to soil, through microbial decomposition. There are many ways to intervene in forests, including logging and planting, which usually affect soil nutrients. Cutting timber can heavily impact soil compaction, temperature and diurnal fluctuation, causing changes in soil (Camenzind et al., 2018; DeLuca and Aplet, 2008). Excessive cutting may result in serious consequences, such as forest degradation or soil erosion. In contrast, appropriate harvesting promotes soil nutrients through complex microbial decomposition (Jamroz et al., 2014; Ma et al., 2013; Zhou et al., 2015).

- 30 Many have done related work to reveal the relationship between timber cutting and forest soil (Guan et al., 2018; Gao et al., 2013). Recently, many researchers have attached great importance to understanding the impacts of cutting and soil (Arevalo-Gardini et al., 2015; Pang et al., 2011). Some studies have shown that harvesting not only deteriorate the physical properties of the soil, especially the soil water holding capacity and soil porosity, but also affects the chemical properties. (Caldato et al., 2016; Parfitt et al., 2014; Yang et al., 2016); the soil bulk density was increasing and soil was being eroded. (Gerke and Hierold,
- 35 2012; Hieke and Schmidt, 2013; Zhou et al., 2010); and organic matter, N (nitrogen), P (phosphorus), K (potassium), and other minerals were also reduced after cutting. (Ikurekong and Akpabio, 2005; Ong et al., 2012; Ozcan and Gokbulak, 2015). Some others have found the change of forest stand structure after lumbering (De Nicola et al., 2017; Oyen and Nilsen, 2004; Zhirin and Knyazeva, 2012). Forest ecosystems can be disturbed by cutting heavily. Many trees have been taken away, and the composition of tree species also becomes different, such as the changes of dominant of tree species and spatial distribution
- 40 structure. Some scientists have also tried to determine the effects of timber harvesting on biodiversity (Barna and Bosela, 2015; Dechene and Buddle, 2009; Okonogi and Fukuda, 2017). These studies have explored that high-intensity interference may adversely affect biodiversity, while low-intensity interference may benefit biodiversity for a long time. However, most of these studies focused on the short- or medium-term effects of plantation and timber harvesting, in part because of the lack of longterm data. In addition, most of them analyse the variability of individual property rather than from the perspective of the overall
- 45 properties. Therefore, it is necessary to reveal the effects of wood harvesting in mixed forests over a longer period of time from the perspective of overall properties. The study aims to describe the impacts of timber cutting on selected physical and chemical soil properties throughout a nine year period in the broadleaf mixed forest located in the Daxing'anling mountain range, Northeast China. We try to reveal the change of soil quality in 3, 6 and 9 years after cutting in different width of strip on both physical and chemical properties. Focusing on the impacts of cutting strip reforms, we would like to enrich the existing
- 50 literature basing on the impacts of cutting strip reforms, which is centred on cutting intensity in forest plantations over a certain time.

In addition, we use fuzzy mathematics and multivariate statistical analyses (such as PCA) to calculate the comprehensive index of combining soil physical with chemical properties and evaluate its soil quality by the value. Since cutting also changes several soil properties simultaneously and these properties usually interact mutually, it is crucial to collectively reflect the

[revised manuscript text omitted]

- 100 and 20 cm were gathered as sample and they were mixed directly. The sampling was implemented based on the national standard for gathering and handling soil samples in forest (Zheng et al., 2008). Soil samples to texting the soil properties are taken from the A, B, and C sections in each plot, with 5 samplings of each subplots. In order to test physical properties, undisturbed soil samples were held in their initial shapes by placing them into aluminium boxes to prevent them from being squeezed and becoming deformed. In order to analyse chemical properties of soil samples, disturbed samples were put inside
- 105 plastic bags, sealed and labelled. Three soil samples from the A, B, and C sections in each plot were evenly mixed and airdried and finally 5 samples per treatment.

The soil physical properties analysed here selected soil bulk density, soil maximum water-holding capacity, soil capillary water-holding capacity, soil non-capillary porosity, soil capillary porosity and soil total porosity. Soil bulk density is closely related to soil porosity, and is one of the important indicators reflecting soil physical properties. Soil bulk density is related to

- 110 the development of soil and can reflect the permeability and water permeability of soil. Current studies show that soil bulk density is related to the compactness of soil. The smaller the soil bulk density, the looser the soil will be, which means there are more aggregates in the soil and the stronger the ability of water conservation of soil is. Soil porosity is also an important indicator of soil physical properties. Water, nutrients and air in soil are stored in soil pore. Among them, capillary porosity is particularly important. Most of the available water in soil is stored in the capillary porosity of soil. The larger the capillary
- 115 porosity, the higher the content of available water stored in soil, thus providing more water for plant survival and promoting vegetation growth. Soil non-capillary porosity is related to soil permeability. The higher the non-capillary porosity is, the faster the infiltration rate of precipitation is, and the stronger the ability of water conservation and soil and water conservation is. Soil water-holding capacity is an important index reflecting soil hydrological performance and water conservation capacity of forest. The stronger water-holding capacity, the more water can be stored in soil, the more precipitation can be intercepted. To
- 120 a certain extent, it can help to avoid the erosion and loss of soil and water. So because of the function of these indicators, we choose them to reflect the influence of year and strip to see if the erosion of soil and water can be managed. Meanwhile, indicators of soil chemical properties commonly, such as organic matter, total nitrogen (N), total phosphorus (P), total

potassium (K), water-soluble nitrogen (N), rapidly avaliable phosphrous (P), and rapidly avaliable potassium (K) were also considered in this study.

- 125 According to the national standard/protocol, analyses of soil physical and chemical properties were done. (Zhang et al., 1984). The water holding capacity was analysed with the cutting ring method (LY/T1215-1999) (Forestry); organic matter was quantified with the potassium dichromate oxidation-external heating method (LY/T 1237-1999) (Forestry); total nitrogen was assessed via the perchloric acid-sulfuric acid digestion diffusion absorption method (LY/T 1228-1999) (Forestry); watersoluble nitrogen was extracted with the alkaline hydrolysis-diffusion absorption method (LY/T 1229-1999) (Forestry); total
- 130 phosphorus was estimated with the perchloric acid-sulfuric acid-soluble Mo-Sb colorimetry method (LY/T 1232-1999) (Forestry); rapidly available phosphorus was gauged with the hydrochloric acid-ammonium fluoride extraction method (LY/T 1233-1999) (Forestry); total potassium was measured with the sodium hydroxide alkali fusion-flame photometry method (LY/T 1234-1999) (Forestry); and rapidly available potassium was tested with the ammonium acetate extraction-flame photometry method (LY/T 1236-1999) (Forestry).

**135 2.4 Data analyses**

The soil testing results got finally are the representative for the average value from the A, B, and C sections in each strip relatively. With the data derived from laboratory experiment and pre-processing, the change in soil physical and chemical properties was calculated under different cutting strip widths and different year by repeated measures ANOVA using SPSS. Because repeated measures is a term used when the same entities take part in all conditions of an experiments. In our study,

- 140 we focus on same subplots testing for different years and strips so that it fits for the method. Besides, two-way repeated measures ANOVA was selected because two independent variables have been manipulated in the experiments, which are year and strip. It is appropriate because each plot does all of the conditions in the experiment, and provides a score for each permutation of two variables. Firstly, SPSS produces a text that look at whether the data have violated the assumption of sphericity. According to the Mauchly's test for these data, where the significance value is most important, if the value is less
- 145 than 0.05, we must accept the hypothesis that the variances of the difference between levels were significantly different and a test statistics (F-ratio) that simply cannot be compared to tabulated values of the F-distribution, which means it needs to be corrected. There are three ways to adjust it in SPSS. The basic way is looking at the Greenhouse-Geisser estimate of sphericity ( $\epsilon$ ) in the SPSS handout. When  $\epsilon$ >0.75 then use the Huynh-Feldt correction. When  $\epsilon$ <0.75 then use the Greenhouse-Geisser correction. The specific results of the sphericity were not shown here but we adopted this method.
- 150 Actually, the aggregate effect of the cutting strip width was the results particularly interested in, which called for a multivariate analysis. Nevertheless, possible correlations among different variables in this model brought statistical complications (Melquiades et al., 2013). To overcome this challenge, we adopted fuzzy mathematics and a principal component analysis. Because of different attributes and dimensions of diverse soil quality indicators, they must be processed before soil quality can be comprehensively evaluated. Data standardization is a statistical method to compare different dimensions and different types
- 155 of set of indicators (Fan et al., 2015). In this study, first, the soil physical and chemical properties were standardized and

transformed into dimensionless values between 0 and 1, to normalize the dimensions of the indicators. In data standardization, data are divided into 2 types: positive and negative effects. In this study, except for soil bulk density, the other indicators are positive effects. The positive and negative effects are calculated by Eq. (1) and Eq. (2) respectively. The method follows three principles: the relative difference of data within the same index remains unchanged, the relative difference between different indices remains unchanged, and the maximum value after standardization is equal.

160

$$F(X_i) = (X_{i\max} - X_{ij}) / (X_{i\max} - X_{i\min}),$$
(1)

$$F(X_i) = (X_{ij} - X_{i\min}) / (X_{i\max} - X_{i\min})$$
(2)

That is, we computed  $F(X_i)$ , which is the membership value of soil property *i*, reflecting the evaluation as follows, where  $X_{i\max}$  is the maximum measured value of soil property *i*;  $X_{ij}$  is the average value of the measured sample of soil property *i*;

165 and  $X_{i\min}$  is the minimum measured value of soil property *i*.

Because the importance of each factor is different, namely, the degree of impact on soil quality is diverse, it needs to be given distinct weights. In this study, SPSS is used to analyse the standardized data of 13 indicators using a principal component analysis, and the contribution rate and cumulative contribution rate of each factor are calculated. The load matrix is obtained by common factor rotation, and the common factor variance of the soil quality index is calculated to show its contribution to

- 170 the variation of soil quality belonging to the soil physical and chemical properties. The proportion of the common factor variance of each index to the total common factor variance is taken as the weight of each index. Based on the evaluation factors of membership degree and weight determination, using the weighted method and addition rule in fuzzy mathematics, we use Eq. (3) to calculate the soil quality of different cutting strip widths in these years. *F* is the comprehensive index of soil quality, and *Wi* is the weight of each soil factor, which reflects the importance of each evaluation
- 175 index.

$$F = \sum W_i \times F(X_i) , \qquad (3)$$

**3 Results**

**3.1 Impacts on soil physical properties individually**

180

As shown in Table 1, all of these indicators showed a certain variation, indicating that soil physical properties could be at least influenced over time. However, some of these changes were not significant. Soil bulk density showed a decline from 6 to 9 years after the cutting in most cutting strips while the changes of other indicators seem to more complex. In different years, there were differences in the correlation between the changes of indices and the width of cutting strips. Three years after cutting, soil bulk density decreased and then increased with the increasing of the width of strip. The lowest mean value appeared in the 10m strip, which is 0.62. In 6 and 9 years after cutting, the mean value is decreasing as a whole, but the lowest

- 185 value appeared in the width of 6m (0.57) and 10m (0.59). As for other indicators, the trends were hardly to describe, which have many waves and there was diversity in the turning points. So next, we tried to use repeated measures ANOVA to separate the effects of year and strip. Table 2 shows the results of the ANOVA (with corrected F values). The output was split into sections that referred to each of the effects in the model. Looking at the significance values in the table it was clear that there were significant differences (p<0.05) between various years in the indicators of soil bulk density and soil non-capillary
- 190 porosity, and there are significant differences (p<0.05) between various strips in the indicators of soil maximum water-holding capacity, soil capillary water-holding capacity, soil non-capillary porosity and soil total porosity. As for the interaction between these two variables, there are significant differences (p<0.05) in the indicators of soil capillary water-holding capacity, soil non-capillary porosity and soil capillary porosity.

As shown in Table 3 and Figure 2, the mean effects of indicators that had significant differences about year were displayed.

- 195 The value of soil bulk density was highest in 2012 (3 years after cutting). It had significantly decreased after 6 and 9 years after cutting. The value of soil non-capillary porosity was increased in 2015 and decreased in 2018. Only these two indicators of soil physical properties had significant differences during 9 years after cutting. As shown in Table 4 and Figure 3, the mean effects of indicators that had significant differences about strip were displayed. Except for the strip with width of 14m and 18m, the value of soil maximum water-holding capacity of the others cutting strip was higher than the control plot (non-
- 200 cutting). The value of soil capillary water-holding capacity in control plot was the lowest. The value of soil non-capillary porosity in width of 14m was lowest, but the control plot was highest. The value of soil total porosity was highest in control plot and lowest in the width of 14m. As shown in Table 5 and Figure 4, when it comes to the interaction of year and strip, only three indicators have significant differences between the interactions, which are soil capillary water-holding capacity, soil non-capillary porosity and soil capillary porosity. This effect told us that the profile of ratings across dates of different levels of
- 205 year was different for width of strips, which means the influence of strips was changing during the years after cutting.

| Cutting | Soil Bulk             | Soil Maximum     | Soil Capillary        | Soil Non-        | Soil Capillary   | Soil Total   |
|---------|-----------------------|------------------|-----------------------|------------------|------------------|--------------|
| Strip   | Density               | Water-holding    | Water-holding         | capillary        | Porosity (%)     | Porosity (%) |
| Width   | (g·cm -3 ) | Capacity (%)     | Capacity (%)          | Porosity (%)     |                  |              |
|         |                       |                  | 3 years after cutting | 7                |                  |              |
| 6 m     | 0.63±0.09             | 92.63±18.61      | 81.25±13.47           | 7.41±2.20        | 52.9±8.83        | 60.31±12.15  |
| 10 m    | 0.62±0.13             | 96.56±18.74      | 84.10±10.24           | 7.03±2.31        | $55.45 \pm 8.38$ | 62.48±10.79  |
| 14 m    | 0.66±0.13             | 89.51±16.06      | 79.99±12.57           | 8.22±2.93        | $52.03 \pm 7.58$ | 60.25±10.39  |
| 18 m    | $0.72 \pm 0.21$       | 75.36±18.24      | 58.06±12.56           | 13.13±2.39       | $44.08 \pm 8.80$ | 57.21±7.84   |
| Non-    |                       |                  |                       |                  |                  |              |
| cutting | $0.66 \pm 0.10$       | 94.75±10.38      | 98.88±10.74           | 13.01±2.01       | $53.48 \pm 9.84$ | 66.49±9.64   |
|         |                       |                  | 6 years after cutting | 5                |                  |              |
| 6 m     | $0.57 \pm 0.10$       | 95.44±6.46       | 85.25±13.94           | 20.23±2.95       | 46.38±10.8       | 66.61±12.58  |
| 10 m    | $0.60 \pm 0.13$       | $96.79 \pm 5.48$ | 89.23±11.67           | 16.42±2.66       | 51.37±7.02       | 67.79±10.38  |
| 14 m    | $0.65 \pm 0.14$       | 80.15±16.67      | 83.12±9.90            | 9.12±1.56        | 54.81±10.34      | 63.93±12.25  |
| 18 m    | $0.63 \pm 0.17$       | 79.82±19.02      | 78.37±11.30           | $10.58 \pm 2.01$ | 50.11±8.65       | 60.69±12.72  |
| Non-    |                       |                  |                       |                  |                  |              |
| cutting | $0.64{\pm}0.12$       | 96.84±5.59       | 89.64±11.21           | 15.29±2.01       | 49.89±8.89       | 65.18±9.93   |
|         |                       |                  | 9 years after cutting | 5                |                  |              |
| 6 m     | 0.61±0.12             | 98.06±4.33       | 85.21±15.67           | 13.97±1.65       | 52.47±7.85       | 66.44±8.86   |
| 10 m    | $0.59{\pm}0.13$       | $97.65 \pm 5.26$ | 88.47±14.82           | 11.29±2.42       | $54.13 \pm 8.08$ | 65.42±12.32  |
|         |                       |                  |                       |                  |                  |              |

**Table 1. Soil physical properties in 3, 6 and 9 years after cutting**

| 14 m    | $0.62 \pm 0.12$ | 86.08±15.50      | 81.26±12.82 | $9.87 \pm 2.00$  | 53.14±9.98 | 63.01±9.37  |
|---------|-----------------|------------------|-------------|------------------|------------|-------------|
| 18 m    | $0.69{\pm}0.16$ | 79.68±17.90      | 80.12±13.06 | $10.13 \pm 1.72$ | 48.35±9.62 | 58.48±12.07 |
| Non-    |                 |                  |             |                  |            |             |
| cutting | 0.63±0.12       | $94.04 \pm 8.24$ | 91.02±13.45 | 13.51±2.31       | 51.23±9.39 | 64.74±8.99  |

Note: The number in the table is "average  $\pm$  standard deviation". Standard deviation is between strips of all years.

|              | Table 2. Te            | sts of within-subjects
Type III Sum of | effects for so   | Mean   | perties |      |  |  |
|--------------|------------------------|-------------------------------------------|------------------|--------|---------|------|--|--|
| S            | ource                  | Squares                                   | df               | Square | F       | Sig. |  |  |
|              | Soil Bulk Density      |                                           |                  |        |         |      |  |  |
| Year         | Sphericity
Assumed  | 0.02                                      | 2.00             | 0.01   | 9.82    | 0.01 |  |  |
| Strip        | Sphericity
Assumed  | 0.06                                      | 4.00             | 0.02   | 2.97    | 0.05 |  |  |
| Year * Strip | Sphericity
Assumed  | 0.02                                      | 8.00             | 0.00   | 0.72    | 0.67 |  |  |
|              |                        | Soil Maximum Wa                           | ter-holding Ca   | pacity |         |      |  |  |
| Year         | Sphericity
Assumed  | 28.99                                     | 2.00             | 14.50  | 0.16    | 0.85 |  |  |
| Strip        | Sphericity
Assumed  | 3968.03                                   | 4.00             | 992.01 | 7.63    | 0.00 |  |  |
| Year * Strip | Sphericity
Assumed  | 357.75                                    | 8.00             | 44.72  | 0.66    | 0.72 |  |  |
|              |                        | Soil Capillary Wat                        | ter-holding Cap  | pacity |         |      |  |  |
| Year         | Sphericity
Assumed  | 423.98                                    | 2.00             | 211.99 | 2.49    | 0.15 |  |  |
| Strip        | Sphericity
Assumed  | 3013.22                                   | 4.00             | 753.30 | 17.24   | 0.00 |  |  |
| Year * Strip | Sphericity
Assumed  | 1263.39                                   | 8.00             | 157.92 | 5.78    | 0.00 |  |  |
|              |                        | Soil Non-cap                              | oillary Porosity |        |         |      |  |  |
| Year         | Greenhouse-
Geisser | 262.23                                    | 1.00             | 261.07 | 36.76   | 0.00 |  |  |
| Strip        | Sphericity
Assumed  | 247.70                                    | 4.00             | 61.93  | 38.35   | 0.00 |  |  |
| Year * Strip | Sphericity
Assumed  | 417.14                                    | 8.00             | 52.14  | 26.83   | 0.00 |  |  |
|              |                        | Soil Capill                               | lary Porosity    |        |         |      |  |  |
| Year         | Greenhouse-
Geisser | 25.52                                     | 1.02             | 25.03  | 0.73    | 0.44 |  |  |
| Strip        | Sphericity
Assumed  | 368.02                                    | 4.00             | 92.00  | 2.17    | 0.12 |  |  |
| Year * Strip | Sphericity
Assumed  | 299.45                                    | 8.00             | 37.43  | 2.46    | 0.03 |  |  |
|              |                        | Soil Tota                                 | al Porosity      |        |         |      |  |  |
| Year         | Sphericity
Assumed  | 157.00                                    | 2.00             | 78.50  | 1.66    | 0.25 |  |  |
| Strip        | Sphericity
Assumed  | 463.28                                    | 4.00             | 115.82 | 4.89    | 0.01 |  |  |

| Year * Strip | Sphericity
Assumed          | 118.56           | 8.0               | 00 14.           | 82 0.62           | 0.76       |
|--------------|--------------------------------|------------------|-------------------|------------------|-------------------|------------|
| Table 3      | . E stimates of p       | hysical indicate | ors which hav     | e significant d  | ifference in vari | ous years  |
|              | Year                           | Mean             | Std. Error        | 95% confid       | lence Interval    |            |
|              |                                |                  |                   | Lower
Bound   | Upper
Bound    |            |
|              |                                |                  |                   |                  |                   |            |
|              |                                |                  | Soil Bulk Densi   | ty               |                   |            |
|              | 2012                           | 0.66             | 0.05              | 0.51             | 0.81              |            |
|              | 2015                           | 0.62             | 0.06              | 0.46             | 0.77              |            |
|              | 2018                           | 0.63             | 0.06              | 0.47             | 0.78              |            |
|              |                                | Soil             | Non-capillary Po  | orosity          |                   |            |
|              | 2012                           | 9.76             | 0.94              | 7.15             | 12.37             |            |
|              | 2015                           | 14.33            | 0.82              | 12.05            | 16.61             |            |
|              | 2018                           | 11.75            | 0.77              | 9.62             | 13.88             |            |
| Table 4      | . E <mark>stimates of p</mark> | hysical indicato | ors which have    | e significant di | ifference in vari | ous strips |
|              | Strip                          | Mean             | Std. Error        | 95% confiden     | ce Interval       |            |
|              |                                |                  |                   | Lower
Bound   | Upper Bound       |            |
|              |                                | Soil Maxin       | num Water-hold    | ing Capacity     |                   |            |
|              | 6m                             | 95.38            | 4.07              | 84.09            | 106.67            |            |
|              | 10m                            | 97.00            | 3.97              | 85.98            | 108.02            |            |
|              | 14m                            | 85.25            | 6.73              | 66.57            | 103.92            |            |
|              | 18m                            | 78.29            | 7.21              | 58.27            | 98.31             |            |
|              | CK                             | 95.21            | 3.19              | 86.36            | 104.06            |            |
|              |                                | Soil Capil       | lary Water-holdi  | ng Capacity      |                   |            |
|              | 6m                             | 83.42            | 5.99              | 66.79            | 100.05            |            |
|              | 10m                            | 86.26            | 5.40              | 71.26            | 101.26            |            |
|              | 14m                            | 81.43            | 5.38              | 66.48            | 96.37             |            |
|              | 18m                            | 72.18            | 5.79              | 56.11            | 88.26             |            |
|              | CK                             | 91.40            | 4.98              | 77.58            | 105.22            |            |
|              |                                | Soil             | Non-capillary Po  | orosity          |                   |            |
|              | 6m                             | 13.87            | 0.97              | 11.17            | 16.57             |            |
|              | 10m                            | 11.58            | 0.82              | 9.31             | 13.85             |            |
|              | 14m                            | 9.07             | 0.84              | 6.75             | 11.40             |            |
|              | 18m                            | 11.28            | 0.67              | 9.42             | 13.14             |            |
|              | СК                             | 13.94            | 0.88              | 11.51            | 16.37             |            |
|              |                                | 5                | Soil Total Porosi | ty               |                   |            |
|              | 6m                             | 64.45            | 4.64              | 51.57            | 77.34             |            |
|              | 10m                            | 65.23            | 3.99              | 54.16            | 76.30             |            |
|              | 14m                            | 62.40            | 4.54              | 49.80            | 75.00             |            |

| 18m | 58.79 | 4.72 | 45.68 | 71.91 |
|-----|-------|------|-------|-------|
| CK  | 65.47 | 4.12 | 54.02 | 76.92 |

**Table 5. Estimates of physical indicators which have significant difference in the interaction of year and strip**

| Year * | Strip | Mean              | Std. Error        | 95% confidenc  | e Interval     |
|--------|-------|-------------------|-------------------|----------------|----------------|
|        |       |                   |                   | Lower
Bound | Upper
Bound |
|        |       | Soil Capillary Wa | ater-holding Capa | ncity          |                |
| 2012   | 6m    | 81.25             | 6.73              | 62.55          | 99.95          |
|        | 10m   | 84.10             | 5.12              | 69.88          | 98.32          |
|        | 14m   | 79.99             | 6.29              | 62.54          | 97.44          |
|        | 18m   | 58.06             | 6.28              | 40.63          | 75.49          |
|        | СК    | 94.53             | 3.13              | 85.84          | 103.22         |
| 2015   | 6m    | 84.89             | 6.77              | 66.10          | 103.69         |
|        | 10m   | 88.40             | 5.35              | 73.56          | 103.25         |
|        | 14m   | 83.12             | 4.95              | 69.37          | 96.87          |
|        | 18m   | 78.37             | 5.65              | 62.69          | 94.05          |
|        | СК    | 89.36             | 5.47              | 74.19          | 104.54         |
| 2018   | 6m    | 84.13             | 7.19              | 64.17          | 104.08         |
|        | 10m   | 86.28             | 6.17              | 69.15          | 103.41         |
|        | 14m   | 81.17             | 6.34              | 63.56          | 98.77          |
|        | 18m   | 80.12             | 6.53              | 62.00          | 98.25          |
|        | СК    | 90.32             | 6.43              | 72.47          | 108.17         |
|        |       | Soil Non-ca       | pillary Porosity  |                |                |
| 2012   | 6m    | 7.41              | 0.98              | 4.68           | 10.14          |
|        | 10m   | 7.03              | 1.03              | 4.16           | 9.90           |
|        | 14m   | 8.22              | 1.31              | 4.58           | 11.86          |
|        | 18m   | 13.13             | 1.07              | 10.17          | 16.09          |
|        | СК    | 13.01             | 0.90              | 10.52          | 15.50          |
| 2015   | 6m    | 20.23             | 1.32              | 16.56          | 23.90          |
|        | 10m   | 16.42             | 1.19              | 13.11          | 19.73          |
|        | 14m   | 9.12              | 0.70              | 7.19           | 11.05          |
|        | 18m   | 10.58             | 0.90              | 8.09           | 13.07          |
|        | СК    | 15.29             | 0.90              | 12.79          | 17.79          |
| 2018   | 6m    | 13.97             | 0.74              | 11.93          | 16.02          |
|        | 10m   | 11.29             | 1.08              | 8.29           | 14.29          |
|        | 14m   | 9.87              | 0.90              | 7.39           | 12.35          |
|        | 18m   | 10.13             | 0.77              | 7.99           | 12.27          |
|        | СК    | 13.51             | 1.03              | 10.64          | 16.38          |
|        |       | Soil Coni         | llam, Daragity    |                |                |

| 2012 | 6m  | 52.90 | 3.95 | 41.94 | 63.86 |
|------|-----|-------|------|-------|-------|
|      | 10m | 55.45 | 3.75 | 45.05 | 65.86 |
|      | 14m | 52.03 | 3.39 | 42.62 | 61.44 |
|      | 18m | 44.08 | 3.93 | 33.16 | 55.00 |
|      | CK  | 53.48 | 4.40 | 41.27 | 65.69 |
| 2015 | 6m  | 46.38 | 4.83 | 32.97 | 59.79 |
|      | 10m | 51.37 | 3.14 | 42.65 | 60.09 |
|      | 14m | 54.81 | 4.62 | 41.98 | 67.64 |
|      | 18m | 50.11 | 3.87 | 39.37 | 60.85 |
|      | CK  | 49.89 | 3.98 | 38.85 | 60.93 |
| 2018 | 6m  | 52.47 | 3.51 | 42.73 | 62.21 |
|      | 10m | 54.13 | 3.62 | 44.09 | 64.17 |
|      | 14m | 53.14 | 4.46 | 40.75 | 65.53 |
|      | 18m | 48.35 | 4.30 | 36.41 | 60.29 |
|      | СК  | 51.23 | 4.20 | 39.58 | 62.89 |

**3.2 Impacts on soil chemical properties individually**

As shown in Table 6, the trend of all indicators are not the same either by strip or by year. However, most of their values increased and then decreased with an increase in cutting strip with, which the peak value often was in 10m and 14m. But it seemed to not significant. On the other hand, by the year, the value of most indicators were decreased in 6 years after cutting compared with 3 years and basically remained unchanged in 9 years after cutting. To get more details about the effects of year and strip, the repeated measures ANOVA was applied.

Table 7 shows the results of the ANOVA (with corrected F values). The output was split into sections that referred to each of the effects in the model. Looking at the significance values in the table it was clear that there were significant differences (p<0.05) between various years in the indicators of water-soluble nitrogen and rapidly available phosphorus, and there are significant differences (p<0.05) between various strips in these two indicators. As for the interaction between these two

variables, there are no significant differences (p < 0.05) in all indicators of soil chemical properties.

As shown in Table 8 and Figure 5, the mean effects of indicators of soil chemical properties that had significant differences about year were displayed. The value of water-soluble nitrogen was highest in 2012, which was 524.53. It has significantly decreased after 6 and 9 years after cutting, which is the same as the trend of the value of rapidly available phosphorus. Only these two indicators of soil chemical properties had significant differences during 9 years after cutting. As shown in Table 9 and Figure 6, the mean effects of indicators of soil chemical properties that had significant differences about strip were displayed. The value of water-soluble nitrogen and rapidly available phosphorus in control plot was the lowest, and in width

230 of 10m was highest. However, when it comes to the interaction of year and strip, no indicator of soil chemical properties has

significant differences between the interactions, which told us that the profile of ratings across dates of different levels of year had no difference for width of strips, in other words that means the influence of strips weren't changing during the years after cutting.

|         | Table 6. Soil chemical properties in 3, 6 and 9 years after cutting |                     |                                 |                     |                    |                     |                        |
|---------|---------------------------------------------------------------------|---------------------|---------------------------------|---------------------|--------------------|---------------------|------------------------|
| Cutting | Organic                                                             | Total               | Water-soluble                   | Total               | Rapidly Available  | Total Potassium     | Rapidly                |
| Strip   | Matter (g·kg -1 )                                        | Nitrogen            | Nitrogen (mg·kg -1 ) | Phosphorus          | Phosphorus (mg·kg- | $(g \cdot kg^{-1})$ | Available              |
| Width   |                                                                     | $(g \cdot kg^{-1})$ |                                 | $(g \cdot kg^{-1})$ | 1)                 |                     | Potassium              |
|         |                                                                     |                     |                                 |                     |                    |                     | (mg·kg -1 ) |
|         |                                                                     |                     | 3 ye                            | ears after cutting  |                    |                     |                        |
| 6 m     | 21.25±2.96                                                          | 9.25±2.24           | 519.63±74.93                    | 2.20±0.24           | 14.86±2.34         | 9.24±1.62           | 54.32±8.38             |
| 10 m    | 22.90±3.09                                                          | 9.45±3.20           | 545.01±59.56                    | $2.23 \pm 0.27$     | 16.13±2.65         | 9.36±2.56           | $56.05 \pm 8.04$       |
| 14 m    | 22.19±3.35                                                          | 9.21±3.79           | 557.92±70.33                    | 2.41±0.33           | 16.24±2.91         | 10.21±2.11          | 58.48±9.10             |
| 18 m    | 24.28±3.66                                                          | 9.93±1.90           | 530.28±66.50                    | 2.38±0.26           | 15.86±2.62         | 9.35±1.53           | 58.13±8.14             |
| Non-    |                                                                     |                     |                                 |                     |                    |                     |                        |
| cutting | 20.95±1.89                                                          | 8.58±1.38           | 469.81±67.51                    | 2.13±0.25           | 13.90±1.38         | 9.11±2.37           | 55.87±7.72             |
|         |                                                                     |                     | 6 ye                            | ears after cutting  |                    |                     |                        |
| 6 m     | 21.32±2.62                                                          | 8.82±2.84           | 481.39±55.38                    | $2.08 \pm 0.42$     | 13.42±2.52         | 9.02±1.85           | 53.11±5.90             |
| 10 m    | $21.90 \pm 2.50$                                                    | 9.26±1.71           | 512.36±55.51                    | 2.11±0.27           | 14.97±2.32         | 9.14±1.87           | 57.12±7.63             |
| 14 m    | 22.59±3.17                                                          | 8.91±3.16           | 500.31±60.19                    | 2.24±0.24           | 14.82±2.11         | 9.57±2.41           | 55.51±11.42            |
| 18 m    | 21.53±1.75                                                          | 8.74±2.34           | 492.13±60.18                    | $2.10{\pm}0.44$     | 13.31±1.56         | 8.81±2.33           | 52.57±9.94             |
| Non-    |                                                                     |                     |                                 |                     |                    |                     |                        |
| cutting | 21.61±2.50                                                          | 8.76±2.71           | 468.24±74.78                    | 2.07±0.37           | 13.81±1.90         | 8.86±2.79           | 54.12±10.65            |
|         |                                                                     |                     | 9 ye                            | ears after cutting  |                    |                     |                        |
| 6 m     | 21.85±2.16                                                          | 8.79±2.54           | 485.69±73.46                    | 2.09±0.33           | 13.85±2.52         | 9.01±2.01           | 53.17±7.59             |
| 10 m    | 22.15±2.55                                                          | 9.38±2.01           | 521.31±84.60                    | 2.16±0.35           | 15.23±3.00         | 9.14±2.45           | 58.14±7.31             |
| 14 m    | 21.57±3.19                                                          | 8.98±2.43           | 507.46±72.66                    | 2.31±0.35           | $14.98 \pm 2.09$   | 9.82±2.56           | $55.74 \pm 8.08$       |
| 18 m    | $21.94{\pm}1.88$                                                    | 8.71±2.17           | 497.52±71.20                    | 2.14±0.27           | 13.72±2.24         | 8.85±2.68           | $52.48{\pm}11.80$      |
| Non-    |                                                                     |                     |                                 |                     |                    |                     |                        |
| cutting | 21.38±2.53                                                          | 8.79±2.79           | 471.21±70.50                    | 2.12±0.30           | 13.94±2.68         | 8.92±2.00           | 55.46±6.75             |

235

Note: The number in the table is "average  $\pm$  standard deviation". Standard deviation is between strips of all years.

| Table 7. Tests of within-subjects effects for son chemical properties | Table 7. Tests of withir | 1-subjects effects for | r soil chemical | properties |
|-----------------------------------------------------------------------|--------------------------|------------------------|-----------------|------------|
|-----------------------------------------------------------------------|--------------------------|------------------------|-----------------|------------|

|              | Source             | Type III Sum of Squares | df          | Mean Square | F    | Sig. |
|--------------|--------------------|-------------------------|-------------|-------------|------|------|
|              |                    | Orga                    | anic Matter |             |      |      |
| Year         | Sphericity Assumed | 4.68                    | 2.00        | 2.34        | 0.78 | 0.49 |
| Strip        | Sphericity Assumed | 17.93                   | 4.00        | 4.48        | 0.84 | 0.52 |
| Year * Strip | Sphericity Assumed | 24.88                   | 8.00        | 3.11        | 1.08 | 0.40 |
|              |                    | Tota                    | al Nitrogen |             |      |      |
| Year         | Greenhouse-Geisser | 2.29                    | 1.04        | 2.21        | 1.69 | 0.26 |

| Strip        | Sphericity Assumed | 3.43                 | 4.00                  | 0.86                 | 0.12            | 0.97 |
|--------------|--------------------|----------------------|-----------------------|----------------------|-----------------|------|
| Year * Strip | Sphericity Assumed | 3.68                 | 8.00                  | 0.46                 | 0.21            | 0.99 |
|              |                    | Wat                  | er-soluble Nitrogen   |                      |                 |      |
| Year         | Sphericity Assumed | 16191.40             | 2.00                  | 8095.70              | 10.21           | 0.01 |
| Strip        | Sphericity Assumed | 30978.00             | 4.00                  | 7744.50              | 8.64            | 0.00 |
| Year * Strip | Sphericity Assumed | 5188.12              | 8.00                  | 648.52               | 0.35            | 0.94 |
|              |                    | Т                    | otal Phosphorus       |                      |                 |      |
| Year         | Sphericity Assumed | 0.30                 | 2.00                  | 0.15                 | 3.98            | 0.06 |
| Strip        | Sphericity Assumed | 0.44                 | 4.00                  | 0.11                 | 2.36            | 0.10 |
| Year * Strip | Sphericity Assumed | 0.10                 | 8.00                  | 0.01                 | 0.54            | 0.82 |
|              |                    | Rapidly              | Available Phosphoru   | 15                   |                 |      |
| Year         | Sphericity Assumed | 24.69                | 2.00                  | 12.34                | 30.62           | 0.00 |
| Strip        | Sphericity Assumed | 32.76                | 4.00                  | 8.19                 | 11.30           | 0.00 |
| Year * Strip | Sphericity Assumed | 9.33                 | 8.00                  | 1.17                 | 0.41            | 0.91 |
|              |                    |                      | Fotal Potassium       |                      |                 |      |
| Year         | Greenhouse-Geisser | 1.98                 | 1.04                  | 1.91                 | 0.26            | 0.64 |
| Strip        | Sphericity Assumed | 8.22                 | 4.00                  | 2.05                 | 1.33            | 0.30 |
| Year * Strip | Sphericity Assumed | 0.46                 | 8.00                  | 0.06                 | 0.05            | 1.00 |
|              |                    | Rapidly              | y Available Potassiun | n                    |                 |      |
| Year         | Sphericity Assumed | 58.97                | 2.00                  | 29.49                | 0.82            | 0.47 |
| Strip        | Sphericity Assumed | 132.52               | 4.00                  | 33.13                | 0.80            | 0.54 |
| Year * Strip | Sphericity Assumed | 97.02                | 8.00                  | 12.13                | 0.59            | 0.78 |
|              | Table 8. Estimates | of chemical indicato | rs which have sign    | ificant difference i | n various years |      |

|
Year | Mean   | Std. Error         | 95% confide | ence Interval |
|----------|--------|--------------------|-------------|---------------|
|          |        |                    | Lower       | Upper         |
|          |        |                    | Bound       | Bound         |
|          | W      | ater-soluble Nitro | gen         |               |
|
2012 | 524.53 | 24.12              | 457.57      | 591.49        |
| 2015     | 490.89 | 25.54              | 419.97      | 561.80        |
| 2018     | 496.64 | 28.86              | 416.50      | 576.78        |
|          |        |                    |             |               |

|                              |                                                                                               | 2012  | 15.40      | 0.79       | 13.20                   | 17.59       |  |  |  |  |  |  |  |
|------------------------------|-----------------------------------------------------------------------------------------------|-------|------------|------------|-------------------------|-------------|--|--|--|--|--|--|--|
|                              |                                                                                               | 2015  | 14.07      | 0.89       | 11.60                   | 16.54       |  |  |  |  |  |  |  |
|                              |                                                                                               | 2018  | 14.34      | 0.95       | 11.70                   | 16.99       |  |  |  |  |  |  |  |
|                              | Table 9. Estimates of chemical indicators which have significant difference in various strips |       |            |            |                         |             |  |  |  |  |  |  |  |
| Strip                        | Strip Mean                                                                                    |       | Std. Error |            | 95% confidence Interval |             |  |  |  |  |  |  |  |
|                              |                                                                                               |       | Lo         | ower Bound |                         | Upper Bound |  |  |  |  |  |  |  |
| Water-soluble Nitrogen       |                                                                                               |       |            |            |                         |             |  |  |  |  |  |  |  |
| 6m                           | 495.57                                                                                        | 25.23 | 3          | 425.53     |                         | 565.61      |  |  |  |  |  |  |  |
| 10m                          | 526.23                                                                                        | 24.18 | 3          | 459.08     |                         | 593.37      |  |  |  |  |  |  |  |
| 14m                          | 521.90                                                                                        | 28.21 |            | 443.56     |                         | 600.23      |  |  |  |  |  |  |  |
| 18m                          | 506.64                                                                                        | 28.24 | ŀ          | 428.25     |                         | 585.04      |  |  |  |  |  |  |  |
| CK                           | 469.75                                                                                        | 27.64 | ļ          | 393.00     |                         | 546.50      |  |  |  |  |  |  |  |
| Rapidly Available Phosphorus |                                                                                               |       |            |            |                         |             |  |  |  |  |  |  |  |
| 6m                           | 14.04                                                                                         | 1.02  |            | 11.21      |                         | 16.88       |  |  |  |  |  |  |  |
| 10m                          | 15.44                                                                                         | 0.89  |            | 12.97      |                         | 17.92       |  |  |  |  |  |  |  |
| 14m                          | 15.35                                                                                         | 0.89  |            | 12.89      |                         | 17.80       |  |  |  |  |  |  |  |
| 18m                          | 14.30                                                                                         | 0.83  |            | 11.98      |                         | 16.61       |  |  |  |  |  |  |  |
| СК                           | 13.88                                                                                         | 0.83  |            | 11.57      |                         | 16.20       |  |  |  |  |  |  |  |
|                              |                                                                                               |       |            |            |                         |             |  |  |  |  |  |  |  |

**240 **3.3 Impacts on Soil Physical and Chemical Properties comprehensively**

**3.3.1** Determining the Weights of Indices**

[revised manuscript text omitted]

**4** Discussion**

- 270 The results showed that the effect of bandwidth on the physical properties of the soil surface was more significant than the chemical nature. Schwendenmann, L. (2000) also believed removing vegetation had an effect on the physical soil properties. In our study, four of the six indicators of physical properties showed significant differences in the change of bandwidth, while only two of the chemical properties showed significant differences. This was because the physical properties here were mostly selected as indicators to reflect the capacity of the soil to hold water, and the soil erosion or loss in forests was closely related
- 275 to human disturbances. This was also proved in Borrelli, P. et.al (2017) study said that about half of the soil loss (45.3%) was predicted for the logged areas in Italy. However, in chemical properties, there were only water-soluble nitrogen and rapidly available phosphorus having significant effects within various strips. This showed that the bandwidth harvesting was more affecting the growth of the remaining vegetation, the rate of absorption of elements in the soil changes, and the ionic activity in the soil was intensified. In fact, there have been many studies about it, however, the relationship between soil chemical
- 280 properties and logging in different regions was various especially for the stand age (Schwendenmann, L., 2000). For us, in this stage, the influence of plant available elements effected by cutting of strip was more obvious. What's more, the effect of restoring years after cutting on the physical properties of soil surface seemed to be superior to chemical properties, but this was not supported by special theory, which was directly reflected from the number of indicators. There was no definitive answer of the recovery period to stand disturbances (Zang, R., and Ding, Y, 2009; Griffiths, P. et. al, 2014), but 9 years should not restore forest soil performance unfortunately.
- 285 not restore forest soil performance unfortunately.
- Our results showed that the width of the cutting strip had a significant impact on soil physical and chemical properties comprehensively. In general, the soil bulk density decreases and then increases, but soil porosity and water holding capacity increase and then decrease as width increases after 9 years of cutting reform, echoing the results reported in the literature (Jennings et al., 2012; Lu, 2006; Makineci et al., 2007). Likewise, an increase in the width of the cutting strip, after 9 years of
- 290 recovery in our study, could cause a recovery but then loss of soil nutrients (N, P, and K), which is parallel to the finding of existing studies (XU and WEI, 2013; Ying et al., 2012). In addition to confirming existing findings, our study shed new light on the aggregate impact of cutting strips on both soil physical and chemical properties. The results from PCA revealed that the first principal component was exclusively associated with soil chemical properties, which explained most variation in the impact of cutting strips, and the second principal component was mostly linked to soil physical properties. Therefore, people
- 295 are most concerned about the loss of soil nutrients (especially phosphorus and potassium) due to the excessive width of cutting strips in the forest, which has difficulty recovering in a short time. Without nutrient supplementation, if not fertilized, soil nutrient loss will reduce long-term soil productivity and lead to forest degradation.

[revised manuscript text omitted]

Schwendenmann, L.: Soil properties of boreal riparian plant communities in relation to natural succession and clear-cutting,

Peace River lowlands, Wood Buffalo National Park, Canada, Water, air, and soil pollution, 122(3-4), 449-467, 2000.

**415 XU, Q., and WEI, X. J. F. R. M.: Effects of the Structure Readjustment of Mixed Stands of Ash and Larch on Physical and Chemical Properties of Soil in Ash Stand, 12, 2013. Yang, J., Xu, X. L., Liu, M. X., Xu, C. H., Luo, W., Song, T. Q., Du, H., and Kiely, G.: Effects of Napier grass management**

on soil hydrologic functions in a karst landscape, southwestern China, Soil & Tillage Research, 157, 83-92, 10.1016/j.still.2015.11.012, 2016.

 Ying, M., Zhou, L., and Yin, W. J. S. S. S.: Effects of Different Thinning Manners on the Soil Organic Carbon Content of Larix olgensis Plantations, 48, 170-173, 2012.
 Zang, R., and Ding, Y: Forest recovery on abandoned logging roads in a tropical montane rain forest of Hainan Island, China, Acta Oecologica, 35(3), 462-470, 2009.
 Zhang, W.R. Methods of Soil Location Study in Forestry; Forestry Publishing House: Beijing, China, 1984.

425 Zheng, L.F., Zhou, X.N., Wu, Z.L., Luo, C.J., Cai, R.T. and Lin, H.M. Analysis on soil physic-chemical properties of a natural

forest 10 years after high intensity cutting. For. Res., 21, 106-109, 2018(in Chinese)

Zhirin, V. M., and Knyazeva, S. V.: Changes in the forest cover after intense logging in southern taiga of the russian federation, Contemporary Problems of Ecology, 5, 669-676, 10.1134/s1995425512070104, 2012. Zhou, X. N., Zhou, Y., Zhou, C. J., Wu, Z. L., Zheng, L. F., Hu, X. S., Chen, H. X., and Gan, J. B.: Effects of Cutting

430 Intensity on Soil Physical and Chemical Properties in a Mixed Natural Forest in Southeastern China, Forests, 6, 4495-4509, 10.3390/f6124383, 2015.

Zhou, Z. C., Gan, Z. T., Shangguan, Z. P., and Dong, Z. B.: Effects of grazing on soil physical properties and soil erodibility in semiarid grassland of the Northern Loess Plateau (China), Catena, 82, 87-91, 10.1016/j.catena.2010.05.005, 2010.

---

## Author Comment (AC2) · 2 Aug 2019

Thanks for paying time to this manuscript. I have read the suggestions carefully and tried my best to modify it. I hope you can feel it having improvement. I have rewritten the introduction section to make the objectives clearly. And for the overall soil properties, I have changed the name to "soil physical and chemical properties comprehensively". I think you are right that it is not really overall properties. Next, the material and methods parts also be corrected. I think I have describe the method of sampling clearly. And I also described the parameters I selected as physical properties. What's more, I take a new method to describe the difference between strip

and year separately. I hope it can be better understood. Last, the discussion part was reorganized structure and try to get a deeper analysis. Best wishes.

Please also note the supplement to this comment:
https://www.soil-discuss.net/soil-2019-10/soil-2019-10-AC2-supplement.pdf

---

## Author Comment (AC3) · 2 Aug 2019

I'm sorry for that the manuscript give you a bad impression. I have used another statistic method to try to describe the effects of strip and year and used some graphs to replace the tables. As for the mistakes in English, I apologize to it and I admit there are some limitations for me in using English to write a thesis. What's more, I didn't pay attention to the details about the font size and layout before. I'm ashamed to give this form of paper to you. I hope you can see the new version.

Please also note the supplement to this comment:

[Figure]

https://www.soil-discuss.net/soil-2019-10/soil-2019-10-AC3-supplement.pdf

[Figure]

**Supplement:**

**Changes in soil properties in a low-quality broadleaf mixed forest after cutting strip reforms in a 9-year period in Northeastern China**

Huiwen Guan, Xibin Dong , Tian Zhang, Yuan Meng, Jiafu Ruan, Zhiyong Wang

Key Laboratory of Sustainable Forest Management and Environmental Microorganism Engineering of Heilongjiang Province, Northeast Forestry University, Harbin, 150040, China

*Correspondence to*: Xibin Dong (xibindong@nefu.edu.cn)

**Abstract.** Strip reforms with widths of 6 m, 10 m, 14 m and 18 m were carried out in a low-quality broadleaf mixed forest in Greater Khingan Mountains. The influence of time on soil properties, including physical and chemical properties, were analysed on the basis of data of the soil components obtained from nine consecutive years (from 2010 to 2018). First, use the repeated measures ANOVA to distinguish the effects of various width of cutting strip and years. In the meantime, a principal component analysis was used to determine the weight of each soil indicator, and the fuzzy comprehensive index method was applied to provide further insight into the variation of soil quality. We found that most soil physical properties can be affected by strips while only two indicators can be affected by years. And half of the indicators can be affected by the interaction by strip and year. As for soil chemical properties, only two plant available elements (N and P) can be affected either strip or year. In addition, no indicator was changed by the interaction. Over the 9 years, soil physical properties displayed more differences than chemical properties across cutting strip widths. However, it's not enough for the properties to recover for 9 years unfortunately. In view of the current research years, the soil quality could not be restored in the 18-m harvesting zone within nine years. The cutting width of 10 m is more obvious than that of other transformation widths, so 10 m is the best width for cutting strips for the forest. The study provides reference for the production management of broadleaf mixed forests in the region and other similar areas. A larger width of the cutting strip should be forbidden for this type of forest here. Moreover, for forest soil conditions, we need to continue long-term observations.

**1 Introduction**

In regard to maintaining the productivity and sustainability of forests, soil is a vital factor. For one thing, soil provides the moisture and nutrients to tree growth and supports trees physically. For another, the litter generated by the growing trees can return a great amount of nutrients back to soil, through microbial decomposition. There are many ways to intervene in forests, including logging and planting, which usually affect soil nutrients. Cutting timber can heavily impact soil compaction, temperature and diurnal fluctuation, causing changes in soil (Camenzind et al., 2018; DeLuca and Aplet, 2008). Excessive cutting may result in serious consequences, such as forest degradation or soil erosion. In contrast, appropriate harvesting promotes soil nutrients through complex microbial decomposition (Jamroz et al., 2014; Ma et al., 2013; Zhou et al., 2015).

Many have done related work to reveal the relationship between timber cutting and forest soil (Guan et al., 2018; Gao et al., 2013). Recently, many researchers have attached great importance to understanding the impacts of cutting and soil (Arevalo-Gardini et al., 2015; Pang et al., 2011). Some studies have shown that harvesting not only deteriorate the physical properties of the soil, especially the soil water holding capacity and soil porosity, but also affects the chemical properties. (Caldato et al., 2016; Parfitt et al., 2014; Yang et al., 2016); the soil bulk density was increasing and soil was being eroded. (Gerke and Hierold, 2012; Hieke and Schmidt, 2013; Zhou et al., 2010); and organic matter, N (nitrogen), P (phosphorus), K (potassium), and other minerals were also reduced after cutting. (Ikurekong and Akpabio, 2005; Ong et al., 2012; Ozcan and Gokbulak, 2015).

Some others have found the change of forest stand structure after lumbering (De Nicola et al., 2017; Oyen and Nilsen, 2004; Zhirin and Knyazeva, 2012). Forest ecosystems can be disturbed by cutting heavily. Many trees have been taken away, and the composition of tree species also becomes different, such as the changes of dominant of tree species and spatial distribution structure. Some scientists have also tried to determine the effects of timber harvesting on biodiversity (Barna and Bosela, 2015; Dechene and Buddle, 2009; Okonogi and Fukuda, 2017). These studies have explored that high-intensity interference may adversely affect biodiversity, while low-intensity interference may benefit biodiversity for a long time. However, most of these studies focused on the short- or medium-term effects of plantation and timber harvesting, in part because of the lack of long-term data. In addition, most of them analyse the variability of individual property rather than from the perspective of the overall properties. Therefore, it is necessary to reveal the effects of wood harvesting in mixed forests over a longer period of time from the perspective of overall properties. The study aims to describe the impacts of timber cutting on selected physical and chemical soil properties throughout a nine year period in the broadleaf mixed forest located in the Daxing'anling mountain range, Northeast China. We try to reveal the change of soil quality in 3, 6 and 9 years after cutting in different width of strip on both physical and chemical properties. Focusing on the impacts of cutting strip reforms, we would like to enrich the existing literature basing on the impacts of cutting strip reforms, which is centred on cutting intensity in forest plantations over a certain time.

In addition, we use fuzzy mathematics and multivariate statistical analyses (such as PCA) to calculate the comprehensive index of combining soil physical with chemical properties and evaluate its soil quality by the value. Since cutting also changes several soil properties simultaneously and these properties usually interact mutually, it is crucial to collectively reflect the aggregation effect. At last, we explored the effects of various width of strips in broad-leaved mixed forests ranging from 0 m to 18 m with clear cutting. Consequently, our results and conclusion can help determine the optimal width of the cutting strip for the forests in the region. At the same time, our finding can also benefit other regions with similar forests given the geographic spread.

**2 Study area and methods**

 **2.1 Study area**

The study area was set on the Yuejin Forest Centre, Jiagedaqi Forestry Bureau, Heilongjiang Province, Northeastern China (124°23'48″-124°24'35″E, 50°34'9″-50°34'32″N). The research plots were established in compartment 174. The elevation of the site ranges from 429 to 521 m with a slope of 6-10°. This area has a cold temperate land monsoon climate. The mean annual temperature is -1.3°C, and the annual precipitation is 494.8 mm. The frost-free period is approximately 85-130 days. According to United States Department of Agriculture (USDA) soil taxonomy, the soil on the study area is classified as brown earth. The thickness of soil is 15-30 cm.

The main tree species are *Quercus mongolica* Fisch. ex Ledeb., *Populus davidiana* Dode, *Betula dahurica* Pall., and *Betula platyphylla* Suk. Shrub species on the site are dominated by *Rhododendronea*, covering 12% of the area. Underground herbaceous and liana species are dominated by *Cyperus microiria* and *Pyrola dahurica*, respectively, covering 27% of the area.

**2.2 Plot establishment and measurements**

In March 2009, cutting strips were established in the low-quality broadleaf mixed forest with the widths of 6 m (S1), 10 m (S2), 14 m (S3), and 18 m (S4) (Figure 1), which are cutting plots. The length of the transformation zone was 300 m. When cutting the timbers, mature trees were cut down while the coniferous seedlings and rare tree species were preserved. Every cutting strip was divided into three parts (A, B, C) with lengths of 100 m, cultivating *Larix gmelinii* (Rupr.) Kuzen., *Pinus sylvestris* L.var. *mongolica* Litv., *Pinus koraiensis* Sieb. et Zucc., respectively. A, B, C are subplots. In Fig. 1, the blank parts show the harvesting area while the shadow parts show the reserved band with no cutting, and the bandwidth of the reserved band is the same as the bandwidth of the corresponding transformation band at 6 m, 10 m, 14 m and 18 m. The control plot was set up in the same forest without cutting near the transformation zone, with the distance of 20m, nearly having same original stand state as cutting plots (soil texture, slope, species composition, etc.)

The cutting operation consisted of chainsaw cutting, on-site delimbing and bucking, skidding by human shoulder, and collecting and utilizing branches >5 cm in diameter. This logging method is a common practice in the region, and the width is the most important difference between strips. In August 10, 2012 (3 years after the cutting), August 17, 2015 (6 years after cutting) and August 8, 2018 (9 years after the cutting), we measured the characters of the forest, such as the height and diameter at breast height (DBH). And soil was gathered in the subplots of different strips and did the experiment in laboratory. Because of the limitations of technical means and experimental conditions 10 years ago, we only set up a test area in Greater Khingan Mountains. This may lack the necessary sample repetition for the overall situation of the broadleaf mixed forest in the Greater Khingan Mountains. However, this experiment can reflect the soil changes of the current plot to a certain extent, and provide some reference for future research. In order to meet the statistical needs, in other words, to make the sampling point distribution as uniform as possible, we divided each treatment into three parts. In fact, these three parts have been replanted with three

species, but this is not meaningful for this experiment. (In fact, these three subplots may have differences because replanted species. But in this paper, we neglect it because our main purpose is not that and after the mixture, the effects can be neutralized). We simply took soil samples from the three areas for the composite. The average and standard deviation of the soil physical and chemical properties of each treatment plot were obtained by analysing the results of multiple soil samples according to the random sampling of the soil. Unfortunately, we missed pre-cutting data, so we can't compare this with the data of 3, 6 and 9 years after cutting. Therefore, we looked for a non-cutting plot similar to the treatment site conditions and stand composition in the vicinity of the control plot as pre-cutting.

**2.3 Soil sample measurement**

Since the effect of cutting on soil is mostly on surface soil, only the surface soil layers between 0 and 10 cm and between 10 and 20 cm were gathered as sample and they were mixed directly. The sampling was implemented based on the national standard for gathering and handling soil samples in forest (Zheng et al., 2008). Soil samples to texting the soil properties are taken from the A, B, and C sections in each plot, with 5 samplings of each subplots. In order to test physical properties, undisturbed soil samples were held in their initial shapes by placing them into aluminium boxes to prevent them from being squeezed and becoming deformed. In order to analyse chemical properties of soil samples, disturbed samples were put inside plastic bags, sealed and labelled. Three soil samples from the A, B, and C sections in each plot were evenly mixed and air-dried and finally 5 samples per treatment.

The soil physical properties analysed here selected soil bulk density, soil maximum water-holding capacity, soil capillary water-holding capacity, soil non-capillary porosity, soil capillary porosity and soil total porosity. Soil bulk density is closely related to soil porosity, and is one of the important indicators reflecting soil physical properties. Soil bulk density is related to the development of soil and can reflect the permeability and water permeability of soil. Current studies show that soil bulk density is related to the compactness of soil. The smaller the soil bulk density, the looser the soil will be, which means there are more aggregates in the soil and the stronger the ability of water conservation of soil is. Soil porosity is also an important indicator of soil physical properties. Water, nutrients and air in soil are stored in soil pore. Among them, capillary porosity is particularly important. Most of the available water in soil is stored in the capillary porosity of soil. The larger the capillary porosity, the higher the content of available water stored in soil, thus providing more water for plant survival and promoting vegetation growth. Soil non-capillary porosity is related to soil permeability. The higher the non-capillary porosity is, the faster the infiltration rate of precipitation is, and the stronger the ability of water conservation and soil and water conservation is. Soil water-holding capacity is an important index reflecting soil hydrological performance and water conservation capacity of forest. The stronger water-holding capacity, the more water can be stored in soil, the more precipitation can be intercepted. To a certain extent, it can help to avoid the erosion and loss of soil and water. So because of the function of these indicators, we choose them to reflect the influence of year and strip to see if the erosion of soil and water can be managed. Meanwhile, indicators of soil chemical properties commonly, such as organic matter, total nitrogen (N), total phosphorus (P), total

potassium (K), water-soluble nitrogen (N), rapidly avaliable phosphrous (P), and rapidly avaliable potassium (K) were also considered in this study.

125 According to the national standard/protocol, analyses of soil physical and chemical properties were done. (Zhang et al., 1984). The water holding capacity was analysed with the cutting ring method (LY/T1215-1999) (Forestry); organic matter was quantified with the potassium dichromate oxidation-external heating method (LY/T 1237-1999) (Forestry); total nitrogen was assessed via the perchloric acid-sulfuric acid digestion diffusion absorption method (LY/T 1228-1999) (Forestry); water-soluble nitrogen was extracted with the alkaline hydrolysis-diffusion absorption method (LY/T 1229-1999) (Forestry); total

130 phosphorus was estimated with the perchloric acid-sulfuric acid-soluble Mo-Sb colorimetry method (LY/T 1232-1999) (Forestry); rapidly available phosphorus was gauged with the hydrochloric acid-ammonium fluoride extraction method (LY/T 1233-1999) (Forestry); total potassium was measured with the sodium hydroxide alkali fusion-flame photometry method (LY/T 1234-1999) (Forestry); and rapidly available potassium was tested with the ammonium acetate extraction-flame photometry method (LY/T 1236-1999) (Forestry).

135 **2.4 Data analyses**

The soil testing results got finally are the representative for the average value from the A, B, and C sections in each strip relatively. With the data derived from laboratory experiment and pre-processing, the change in soil physical and chemical properties was calculated under different cutting strip widths and different year by repeated measures ANOVA using SPSS. Because repeated measures is a term used when the same entities take part in all conditions of an experiments. In our study,

140 we focus on same subplots testing for different years and strips so that it fits for the method. Besides, two-way repeated measures ANOVA was selected because two independent variables have been manipulated in the experiments, which are year and strip. It is appropriate because each plot does all of the conditions in the experiment, and provides a score for each permutation of two variables. Firstly, SPSS produces a text that look at whether the data have violated the assumption of sphericity. According to the Mauchly's test for these data, where the significance value is most important, if the value is less

145 than 0.05, we must accept the hypothesis that the variances of the difference between levels were significantly different and a test statistics (F-ratio) that simply cannot be compared to tabulated values of the F-distribution, which means it needs to be corrected. There are three ways to adjust it in SPSS. The basic way is looking at the Greenhouse-Geisser estimate of sphericity ($\varepsilon$) in the SPSS handout. When $\varepsilon > 0.75$ then use the Huynh-Feldt correction. When $\varepsilon < 0.75$ then use the Greenhouse-Geisser correction. The specific results of the sphericity were not shown here but we adopted this method.

150 Actually, the aggregate effect of the cutting strip width was the results particularly interested in, which called for a multivariate analysis. Nevertheless, possible correlations among different variables in this model brought statistical complications (Melquiades et al., 2013). To overcome this challenge, we adopted fuzzy mathematics and a principal component analysis. Because of different attributes and dimensions of diverse soil quality indicators, they must be processed before soil quality can be comprehensively evaluated. Data standardization is a statistical method to compare different dimensions and different types

155 of set of indicators (Fan et al., 2015). In this study, first, the soil physical and chemical properties were standardized and

transformed into dimensionless values between 0 and 1, to normalize the dimensions of the indicators. In data standardization, data are divided into 2 types: positive and negative effects. In this study, except for soil bulk density, the other indicators are positive effects. The positive and negative effects are calculated by Eq. (1) and Eq. (2) respectively. The method follows three principles: the relative difference of data within the same index remains unchanged, the relative difference between different indices remains unchanged, and the maximum value after standardization is equal.

$$F(X_i) = (X_{i\max} - X_{ij}) / (X_{i\max} - X_{i\min}) \quad , \tag{1}$$

$$F(X_i) = (X_{ij} - X_{i\min}) / (X_{i\max} - X_{i\min}) \quad , \tag{2}$$

That is, we computed $F(X_i)$, which is the membership value of soil property $i$, reflecting the evaluation as follows, where $X_{i\max}$ is the maximum measured value of soil property $i$; $X_{ij}$ is the average value of the measured sample of soil property $i$; and $X_{i\min}$ is the minimum measured value of soil property $i$.

Because the importance of each factor is different, namely, the degree of impact on soil quality is diverse, it needs to be given distinct weights. In this study, SPSS is used to analyse the standardized data of 13 indicators using a principal component analysis, and the contribution rate and cumulative contribution rate of each factor are calculated. The load matrix is obtained by common factor rotation, and the common factor variance of the soil quality index is calculated to show its contribution to the variation of soil quality belonging to the soil physical and chemical properties. The proportion of the common factor variance of each index to the total common factor variance is taken as the weight of each index.

Based on the evaluation factors of membership degree and weight determination, using the weighted method and addition rule in fuzzy mathematics, we use Eq. (3) to calculate the soil quality of different cutting strip widths in these years. $F$ is the comprehensive index of soil quality, and $W_i$ is the weight of each soil factor, which reflects the importance of each evaluation index.

$$F = \sum W_i \times F(X_i) \quad , \tag{3}$$

**3 Results**

**3.1 Impacts on soil physical properties individually**

As shown in Table 1, all of these indicators showed a certain variation, indicating that soil physical properties could be at least influenced over time. However, some of these changes were not significant. Soil bulk density showed a decline from 6 to 9 years after the cutting in most cutting strips while the changes of other indicators seem to more complex. In different years, there were differences in the correlation between the changes of indices and the width of cutting strips. Three years after cutting, soil bulk density decreased and then increased with the increasing of the width of strip. The lowest mean value appeared in the 10m strip, which is 0.62. In 6 and 9 years after cutting, the mean value is decreasing as a whole, but the lowest

value appeared in the width of 6m (0.57) and 10m (0.59). As for other indicators, the trends were hardly to describe, which have many waves and there was diversity in the turning points. So next, we tried to use repeated measures ANOVA to separate the effects of year and strip. Table 2 shows the results of the ANOVA (with corrected F values). The output was split into sections that referred to each of the effects in the model. Looking at the significance values in the table it was clear that there were significant differences ($p<0.05$) between various years in the indicators of soil bulk density and soil non-capillary porosity, and there are significant differences ($p<0.05$) between various strips in the indicators of soil maximum water-holding capacity, soil capillary water-holding capacity, soil non-capillary porosity and soil total porosity. As for the interaction between these two variables, there are significant differences ($p<0.05$) in the indicators of soil capillary water-holding capacity, soil non-capillary porosity and soil capillary porosity.

As shown in Table 3 and Figure 2, the mean effects of indicators that had significant differences about year were displayed. The value of soil bulk density was highest in 2012 (3 years after cutting). It had significantly decreased after 6 and 9 years after cutting. The value of soil non-capillary porosity was increased in 2015 and decreased in 2018. Only these two indicators of soil physical properties had significant differences during 9 years after cutting. As shown in Table 4 and Figure 3, the mean effects of indicators that had significant differences about strip were displayed. Except for the strip with width of 14m and 18m, the value of soil maximum water-holding capacity of the others cutting strip was higher than the control plot (non-cutting). The value of soil capillary water-holding capacity in control plot was the lowest. The value of soil non-capillary porosity in width of 14m was lowest, but the control plot was highest. The value of soil total porosity was highest in control plot and lowest in the width of 14m. As shown in Table 5 and Figure 4, when it comes to the interaction of year and strip, only three indicators have significant differences between the interactions, which are soil capillary water-holding capacity, soil non-capillary porosity and soil capillary porosity. This effect told us that the profile of ratings across dates of different levels of year was different for width of strips, which means the influence of strips was changing during the years after cutting.

**Table 1. Soil physical properties in 3, 6 and 9 years after cutting**

| Cutting Strip Width | Soil Bulk Density (g·cm⁻³) | Soil Maximum Water-holding Capacity (%) | Soil Capillary Water-holding Capacity (%) | Soil Non-capillary Porosity (%) | Soil Capillary Porosity (%) | Soil Total Porosity (%) |
|---|---|---|---|---|---|---|
| | | | 3 years after cutting | | | |
| 6 m | 0.63±0.09 | 92.63±18.61 | 81.25±13.47 | 7.41±2.20 | 52.9±8.83 | 60.31±12.15 |
| 10 m | 0.62±0.13 | 96.56±18.74 | 84.10±10.24 | 7.03±2.31 | 55.45±8.38 | 62.48±10.79 |
| 14 m | 0.66±0.13 | 89.51±16.06 | 79.99±12.57 | 8.22±2.93 | 52.03±7.58 | 60.25±10.39 |
| 18 m | 0.72±0.21 | 75.36±18.24 | 58.06±12.56 | 13.13±2.39 | 44.08±8.80 | 57.21±7.84 |
| Non-cutting | 0.66±0.10 | 94.75±10.38 | 98.88±10.74 | 13.01±2.01 | 53.48±9.84 | 66.49±9.64 |
| | | | 6 years after cutting | | | |
| 6 m | 0.57±0.10 | 95.44±6.46 | 85.25±13.94 | 20.23±2.95 | 46.38±10.8 | 66.61±12.58 |
| 10 m | 0.60±0.13 | 96.79±5.48 | 89.23±11.67 | 16.42±2.66 | 51.37±7.02 | 67.79±10.38 |
| 14 m | 0.65±0.14 | 80.15±16.67 | 83.12±9.90 | 9.12±1.56 | 54.81±10.34 | 63.93±12.25 |
| 18 m | 0.63±0.17 | 79.82±19.02 | 78.37±11.30 | 10.58±2.01 | 50.11±8.65 | 60.69±12.72 |
| Non-cutting | 0.64±0.12 | 96.84±5.59 | 89.64±11.21 | 15.29±2.01 | 49.89±8.89 | 65.18±9.93 |
| | | | 9 years after cutting | | | |
| 6 m | 0.61±0.12 | 98.06±4.33 | 85.21±15.67 | 13.97±1.65 | 52.47±7.85 | 66.44±8.86 |
| 10 m | 0.59±0.13 | 97.65±5.26 | 88.47±14.82 | 11.29±2.42 | 54.13±8.08 | 65.42±12.32 |

| | | | | | |
|---|---|---|---|---|---|
| 14 m | 0.62±0.12 | 86.08±15.50 | 81.26±12.82 | 9.87±2.00 | 53.14±9.98 | 63.01±9.37 |
| 18 m | 0.69±0.16 | 79.68±17.90 | 80.12±13.06 | 10.13±1.72 | 48.35±9.62 | 58.48±12.07 |
| Non-cutting | 0.63±0.12 | 94.04±8.24 | 91.02±13.45 | 13.51±2.31 | 51.23±9.39 | 64.74±8.99 |

Note: The number in the table is "average ± standard deviation". Standard deviation is between strips of all years.

**Table 2. Tests of within-subjects effects for soil physical properties**

| Source | | Type III Sum of Squares | df | Mean Square | F | Sig. |
|---|---|---|---|---|---|---|
| Soil Bulk Density | | | | | | |
| Year | Sphericity Assumed | 0.02 | 2.00 | 0.01 | 9.82 | 0.01 |
| Strip | Sphericity Assumed | 0.06 | 4.00 | 0.02 | 2.97 | 0.05 |
| Year * Strip | Sphericity Assumed | 0.02 | 8.00 | 0.00 | 0.72 | 0.67 |
| Soil Maximum Water-holding Capacity | | | | | | |
| Year | Sphericity Assumed | 28.99 | 2.00 | 14.50 | 0.16 | 0.85 |
| Strip | Sphericity Assumed | 3968.03 | 4.00 | 992.01 | 7.63 | 0.00 |
| Year * Strip | Sphericity Assumed | 357.75 | 8.00 | 44.72 | 0.66 | 0.72 |
| Soil Capillary Water-holding Capacity | | | | | | |
| Year | Sphericity Assumed | 423.98 | 2.00 | 211.99 | 2.49 | 0.15 |
| Strip | Sphericity Assumed | 3013.22 | 4.00 | 753.30 | 17.24 | 0.00 |
| Year * Strip | Sphericity Assumed | 1263.39 | 8.00 | 157.92 | 5.78 | 0.00 |
| Soil Non-capillary Porosity | | | | | | |
| Year | Greenhouse-Geisser | 262.23 | 1.00 | 261.07 | 36.76 | 0.00 |
| Strip | Sphericity Assumed | 247.70 | 4.00 | 61.93 | 38.35 | 0.00 |
| Year * Strip | Sphericity Assumed | 417.14 | 8.00 | 52.14 | 26.83 | 0.00 |
| Soil Capillary Porosity | | | | | | |
| Year | Greenhouse-Geisser | 25.52 | 1.02 | 25.03 | 0.73 | 0.44 |
| Strip | Sphericity Assumed | 368.02 | 4.00 | 92.00 | 2.17 | 0.12 |
| Year * Strip | Sphericity Assumed | 299.45 | 8.00 | 37.43 | 2.46 | 0.03 |
| Soil Total Porosity | | | | | | |
| Year | Sphericity Assumed | 157.00 | 2.00 | 78.50 | 1.66 | 0.25 |
| Strip | Sphericity Assumed | 463.28 | 4.00 | 115.82 | 4.89 | 0.01 |

| Year * Strip | Sphericity Assumed | 118.56 | 8.00 | 14.82 | 0.62 | 0.76 |
|---|---|---|---|---|---|---|

**Table 3. Estimates of physical indicators which have significant difference in various years**

| Year | Mean | Std. Error | 95% confidence Interval | |
|---|---|---|---|---|
| | | | Lower Bound | Upper Bound |
| Soil Bulk Density | | | | |
| 2012 | 0.66 | 0.05 | 0.51 | 0.81 |
| 2015 | 0.62 | 0.06 | 0.46 | 0.77 |
| 2018 | 0.63 | 0.06 | 0.47 | 0.78 |
| Soil Non-capillary Porosity | | | | |
| 2012 | 9.76 | 0.94 | 7.15 | 12.37 |
| 2015 | 14.33 | 0.82 | 12.05 | 16.61 |
| 2018 | 11.75 | 0.77 | 9.62 | 13.88 |

210

**Table 4. Estimates of physical indicators which have significant difference in various strips**

| Strip | Mean | Std. Error | 95% confidence Interval | |
|---|---|---|---|---|
| | | | Lower Bound | Upper Bound |
| Soil Maximum Water-holding Capacity | | | | |
| 6m | 95.38 | 4.07 | 84.09 | 106.67 |
| 10m | 97.00 | 3.97 | 85.98 | 108.02 |
| 14m | 85.25 | 6.73 | 66.57 | 103.92 |
| 18m | 78.29 | 7.21 | 58.27 | 98.31 |
| CK | 95.21 | 3.19 | 86.36 | 104.06 |
| Soil Capillary Water-holding Capacity | | | | |
| 6m | 83.42 | 5.99 | 66.79 | 100.05 |
| 10m | 86.26 | 5.40 | 71.26 | 101.26 |
| 14m | 81.43 | 5.38 | 66.48 | 96.37 |
| 18m | 72.18 | 5.79 | 56.11 | 88.26 |
| CK | 91.40 | 4.98 | 77.58 | 105.22 |
| Soil Non-capillary Porosity | | | | |
| 6m | 13.87 | 0.97 | 11.17 | 16.57 |
| 10m | 11.58 | 0.82 | 9.31 | 13.85 |
| 14m | 9.07 | 0.84 | 6.75 | 11.40 |
| 18m | 11.28 | 0.67 | 9.42 | 13.14 |
| CK | 13.94 | 0.88 | 11.51 | 16.37 |
| Soil Total Porosity | | | | |
| 6m | 64.45 | 4.64 | 51.57 | 77.34 |
| 10m | 65.23 | 3.99 | 54.16 | 76.30 |
| 14m | 62.40 | 4.54 | 49.80 | 75.00 |

| | | | | | |
|---|---|---|---|---|---|
| 18m | 58.79 | 4.72 | 45.68 | 71.91 | |
| CK | 65.47 | 4.12 | 54.02 | 76.92 | |

**Table 5. Estimates of physical indicators which have significant difference in the interaction of year and strip**

| Year * Strip | | Mean | Std. Error | 95% confidence Interval | |
|---|---|---|---|---|---|
| | | | | Lower Bound | Upper Bound |
| Soil Capillary Water-holding Capacity | | | | | |
| 2012 | 6m | 81.25 | 6.73 | 62.55 | 99.95 |
| | 10m | 84.10 | 5.12 | 69.88 | 98.32 |
| | 14m | 79.99 | 6.29 | 62.54 | 97.44 |
| | 18m | 58.06 | 6.28 | 40.63 | 75.49 |
| | CK | 94.53 | 3.13 | 85.84 | 103.22 |
| 2015 | 6m | 84.89 | 6.77 | 66.10 | 103.69 |
| | 10m | 88.40 | 5.35 | 73.56 | 103.25 |
| | 14m | 83.12 | 4.95 | 69.37 | 96.87 |
| | 18m | 78.37 | 5.65 | 62.69 | 94.05 |
| | CK | 89.36 | 5.47 | 74.19 | 104.54 |
| 2018 | 6m | 84.13 | 7.19 | 64.17 | 104.08 |
| | 10m | 86.28 | 6.17 | 69.15 | 103.41 |
| | 14m | 81.17 | 6.34 | 63.56 | 98.77 |
| | 18m | 80.12 | 6.53 | 62.00 | 98.25 |
| | CK | 90.32 | 6.43 | 72.47 | 108.17 |
| Soil Non-capillary Porosity | | | | | |
| 2012 | 6m | 7.41 | 0.98 | 4.68 | 10.14 |
| | 10m | 7.03 | 1.03 | 4.16 | 9.90 |
| | 14m | 8.22 | 1.31 | 4.58 | 11.86 |
| | 18m | 13.13 | 1.07 | 10.17 | 16.09 |
| | CK | 13.01 | 0.90 | 10.52 | 15.50 |
| 2015 | 6m | 20.23 | 1.32 | 16.56 | 23.90 |
| | 10m | 16.42 | 1.19 | 13.11 | 19.73 |
| | 14m | 9.12 | 0.70 | 7.19 | 11.05 |
| | 18m | 10.58 | 0.90 | 8.09 | 13.07 |
| | CK | 15.29 | 0.90 | 12.79 | 17.79 |
| 2018 | 6m | 13.97 | 0.74 | 11.93 | 16.02 |
| | 10m | 11.29 | 1.08 | 8.29 | 14.29 |
| | 14m | 9.87 | 0.90 | 7.39 | 12.35 |
| | 18m | 10.13 | 0.77 | 7.99 | 12.27 |
| | CK | 13.51 | 1.03 | 10.64 | 16.38 |
| Soil Capillary Porosity | | | | | |

| | | | | | |
|---|---|---|---|---|---|
| 2012 | 6m | 52.90 | 3.95 | 41.94 | 63.86 |
| | 10m | 55.45 | 3.75 | 45.05 | 65.86 |
| | 14m | 52.03 | 3.39 | 42.62 | 61.44 |
| | 18m | 44.08 | 3.93 | 33.16 | 55.00 |
| | CK | 53.48 | 4.40 | 41.27 | 65.69 |
| 2015 | 6m | 46.38 | 4.83 | 32.97 | 59.79 |
| | 10m | 51.37 | 3.14 | 42.65 | 60.09 |
| | 14m | 54.81 | 4.62 | 41.98 | 67.64 |
| | 18m | 50.11 | 3.87 | 39.37 | 60.85 |
| | CK | 49.89 | 3.98 | 38.85 | 60.93 |
| 2018 | 6m | 52.47 | 3.51 | 42.73 | 62.21 |
| | 10m | 54.13 | 3.62 | 44.09 | 64.17 |
| | 14m | 53.14 | 4.46 | 40.75 | 65.53 |
| | 18m | 48.35 | 4.30 | 36.41 | 60.29 |
| | CK | 51.23 | 4.20 | 39.58 | 62.89 |

**3.2 Impacts on soil chemical properties individually**

As shown in Table 6, the trend of all indicators are not the same either by strip or by year. However, most of their values increased and then decreased with an increase in cutting strip with, which the peak value often was in 10m and 14m. But it seemed to not significant. On the other hand, by the year, the value of most indicators were decreased in 6 years after cutting compared with 3 years and basically remained unchanged in 9 years after cutting. To get more details about the effects of year and strip, the repeated measures ANOVA was applied.

Table 7 shows the results of the ANOVA (with corrected F values). The output was split into sections that referred to each of the effects in the model. Looking at the significance values in the table it was clear that there were significant differences ($p<0.05$) between various years in the indicators of water-soluble nitrogen and rapidly available phosphorus, and there are significant differences ($p<0.05$) between various strips in these two indicators. As for the interaction between these two variables, there are no significant differences ($p<0.05$) in all indicators of soil chemical properties.

As shown in Table 8 and Figure 5, the mean effects of indicators of soil chemical properties that had significant differences about year were displayed. The value of water-soluble nitrogen was highest in 2012, which was 524.53. It has significantly decreased after 6 and 9 years after cutting, which is the same as the trend of the value of rapidly available phosphorus. Only these two indicators of soil chemical properties had significant differences during 9 years after cutting. As shown in Table 9 and Figure 6, the mean effects of indicators of soil chemical properties that had significant differences about strip were displayed. The value of water-soluble nitrogen and rapidly available phosphorus in control plot was the lowest, and in width of 10m was highest. However, when it comes to the interaction of year and strip, no indicator of soil chemical properties has

significant differences between the interactions, which told us that the profile of ratings across dates of different levels of year had no difference for width of strips, in other words that means the influence of strips weren't changing during the years after cutting.

**Table 6. Soil chemical properties in 3, 6 and 9 years after cutting**

| Cutting Strip Width | Organic Matter (g·kg⁻¹) | Total Nitrogen (g·kg⁻¹) | Water-soluble Nitrogen (mg·kg⁻¹) | Total Phosphorus (g·kg⁻¹) | Rapidly Available Phosphorus (mg·kg⁻¹) | Total Potassium (g·kg⁻¹) | Rapidly Available Potassium (mg·kg⁻¹) |
|---|---|---|---|---|---|---|---|
| | | | | 3 years after cutting | | | |
| 6 m | 21.25±2.96 | 9.25±2.24 | 519.63±74.93 | 2.20±0.24 | 14.86±2.34 | 9.24±1.62 | 54.32±8.38 |
| 10 m | 22.90±3.09 | 9.45±3.20 | 545.01±59.56 | 2.23±0.27 | 16.13±2.65 | 9.36±2.56 | 56.05±8.04 |
| 14 m | 22.19±3.35 | 9.21±3.79 | 557.92±70.33 | 2.41±0.33 | 16.24±2.91 | 10.21±2.11 | 58.48±9.10 |
| 18 m | 24.28±3.66 | 9.93±1.90 | 530.28±66.50 | 2.38±0.26 | 15.86±2.62 | 9.35±1.53 | 58.13±8.14 |
| Non-cutting | 20.95±1.89 | 8.58±1.38 | 469.81±67.51 | 2.13±0.25 | 13.90±1.38 | 9.11±2.37 | 55.87±7.72 |
| | | | | 6 years after cutting | | | |
| 6 m | 21.32±2.62 | 8.82±2.84 | 481.39±55.38 | 2.08±0.42 | 13.42±2.52 | 9.02±1.85 | 53.11±5.90 |
| 10 m | 21.90±2.50 | 9.26±1.71 | 512.36±55.51 | 2.11±0.27 | 14.97±2.32 | 9.14±1.87 | 57.12±7.63 |
| 14 m | 22.59±3.17 | 8.91±3.16 | 500.31±60.19 | 2.24±0.24 | 14.82±2.11 | 9.57±2.41 | 55.51±11.42 |
| 18 m | 21.53±1.75 | 8.74±2.34 | 492.13±60.18 | 2.10±0.44 | 13.31±1.56 | 8.81±2.33 | 52.57±9.94 |
| Non-cutting | 21.61±2.50 | 8.76±2.71 | 468.24±74.78 | 2.07±0.37 | 13.81±1.90 | 8.86±2.79 | 54.12±10.65 |
| | | | | 9 years after cutting | | | |
| 6 m | 21.85±2.16 | 8.79±2.54 | 485.69±73.46 | 2.09±0.33 | 13.85±2.52 | 9.01±2.01 | 53.17±7.59 |
| 10 m | 22.15±2.55 | 9.38±2.01 | 521.31±84.60 | 2.16±0.35 | 15.23±3.00 | 9.14±2.45 | 58.14±7.31 |
| 14 m | 21.57±3.19 | 8.98±2.43 | 507.46±72.66 | 2.31±0.35 | 14.98±2.09 | 9.82±2.56 | 55.74±8.08 |
| 18 m | 21.94±1.88 | 8.71±2.17 | 497.52±71.20 | 2.14±0.27 | 13.72±2.24 | 8.85±2.68 | 52.48±11.80 |
| Non-cutting | 21.38±2.53 | 8.79±2.79 | 471.21±70.50 | 2.12±0.30 | 13.94±2.68 | 8.92±2.00 | 55.46±6.75 |

Note: The number in the table is "average ± standard deviation". Standard deviation is between strips of all years.

**Table 7. Tests of within-subjects effects for soil chemical properties**

| | Source | Type III Sum of Squares | df | Mean Square | F | Sig. |
|---|---|---|---|---|---|---|
| | | | Organic Matter | | | |
| Year | Sphericity Assumed | 4.68 | 2.00 | 2.34 | 0.78 | 0.49 |
| Strip | Sphericity Assumed | 17.93 | 4.00 | 4.48 | 0.84 | 0.52 |
| Year * Strip | Sphericity Assumed | 24.88 | 8.00 | 3.11 | 1.08 | 0.40 |
| | | | Total Nitrogen | | | |
| Year | Greenhouse-Geisser | 2.29 | 1.04 | 2.21 | 1.69 | 0.26 |

| | | | | | | |
|---|---|---|---|---|---|---|
| Strip | Sphericity Assumed | 3.43 | 4.00 | 0.86 | 0.12 | 0.97 |
| Year * Strip | Sphericity Assumed | 3.68 | 8.00 | 0.46 | 0.21 | 0.99 |

| Water-soluble Nitrogen | | | | | | |
|---|---|---|---|---|---|---|
| Year | Sphericity Assumed | 16191.40 | 2.00 | 8095.70 | 10.21 | 0.01 |
| Strip | Sphericity Assumed | 30978.00 | 4.00 | 7744.50 | 8.64 | 0.00 |
| Year * Strip | Sphericity Assumed | 5188.12 | 8.00 | 648.52 | 0.35 | 0.94 |

| Total Phosphorus | | | | | | |
|---|---|---|---|---|---|---|
| Year | Sphericity Assumed | 0.30 | 2.00 | 0.15 | 3.98 | 0.06 |
| Strip | Sphericity Assumed | 0.44 | 4.00 | 0.11 | 2.36 | 0.10 |
| Year * Strip | Sphericity Assumed | 0.10 | 8.00 | 0.01 | 0.54 | 0.82 |

| Rapidly Available Phosphorus | | | | | | |
|---|---|---|---|---|---|---|
| Year | Sphericity Assumed | 24.69 | 2.00 | 12.34 | 30.62 | 0.00 |
| Strip | Sphericity Assumed | 32.76 | 4.00 | 8.19 | 11.30 | 0.00 |
| Year * Strip | Sphericity Assumed | 9.33 | 8.00 | 1.17 | 0.41 | 0.91 |

| Total Potassium | | | | | | |
|---|---|---|---|---|---|---|
| Year | Greenhouse-Geisser | 1.98 | 1.04 | 1.91 | 0.26 | 0.64 |
| Strip | Sphericity Assumed | 8.22 | 4.00 | 2.05 | 1.33 | 0.30 |
| Year * Strip | Sphericity Assumed | 0.46 | 8.00 | 0.06 | 0.05 | 1.00 |

| Rapidly Available Potassium | | | | | | |
|---|---|---|---|---|---|---|
| Year | Sphericity Assumed | 58.97 | 2.00 | 29.49 | 0.82 | 0.47 |
| Strip | Sphericity Assumed | 132.52 | 4.00 | 33.13 | 0.80 | 0.54 |
| Year * Strip | Sphericity Assumed | 97.02 | 8.00 | 12.13 | 0.59 | 0.78 |

**Table 8. Estimates of chemical indicators which have significant difference in various years**

| Year | Mean | Std. Error | 95% confidence Interval | |
|---|---|---|---|---|
| | | | Lower Bound | Upper Bound |
| Water-soluble Nitrogen | | | | |
| 2012 | 524.53 | 24.12 | 457.57 | 591.49 |
| 2015 | 490.89 | 25.54 | 419.97 | 561.80 |
| 2018 | 496.64 | 28.86 | 416.50 | 576.78 |

| | | | Rapidly Available Phosphorus | |
|---|---|---|---|---|
| 2012 | 15.40 | 0.79 | 13.20 | 17.59 |
| 2015 | 14.07 | 0.89 | 11.60 | 16.54 |
| 2018 | 14.34 | 0.95 | 11.70 | 16.99 |

**Table 9. Estimates of chemical indicators which have significant difference in various strips**

| Strip | Mean | Std. Error | 95% confidence Interval | |
|---|---|---|---|---|
| | | | Lower Bound | Upper Bound |
| | | Water-soluble Nitrogen | | |
| 6m | 495.57 | 25.23 | 425.53 | 565.61 |
| 10m | 526.23 | 24.18 | 459.08 | 593.37 |
| 14m | 521.90 | 28.21 | 443.56 | 600.23 |
| 18m | 506.64 | 28.24 | 428.25 | 585.04 |
| CK | 469.75 | 27.64 | 393.00 | 546.50 |
| | | Rapidly Available Phosphorus | | |
| 6m | 14.04 | 1.02 | 11.21 | 16.88 |
| 10m | 15.44 | 0.89 | 12.97 | 17.92 |
| 14m | 15.35 | 0.89 | 12.89 | 17.80 |
| 18m | 14.30 | 0.83 | 11.98 | 16.61 |
| CK | 13.88 | 0.83 | 11.57 | 16.20 |

**3.3 Impacts on Soil Physical and Chemical Properties comprehensively**

**3.3.1 Determining the Weights of Indices**

[revised manuscript text omitted]

**4 Discussion**

270  The results showed that the effect of bandwidth on the physical properties of the soil surface was more significant than the chemical nature. Schwendenmann, L. (2000) also believed removing vegetation had an effect on the physical soil properties. In our study, four of the six indicators of physical properties showed significant differences in the change of bandwidth, while only two of the chemical properties showed significant differences. This was because the physical properties here were mostly selected as indicators to reflect the capacity of the soil to hold water, and the soil erosion or loss in forests was closely related

275  to human disturbances. This was also proved in Borrelli, P. et.al (2017) study said that about half of the soil loss (45.3%) was predicted for the logged areas in Italy. However, in chemical properties, there were only water-soluble nitrogen and rapidly available phosphorus having significant effects within various strips. This showed that the bandwidth harvesting was more affecting the growth of the remaining vegetation, the rate of absorption of elements in the soil changes, and the ionic activity in the soil was intensified. In fact, there have been many studies about it, however, the relationship between soil chemical

280  properties and logging in different regions was various especially for the stand age (Schwendenmann, L., 2000). For us, in this stage, the influence of plant available elements effected by cutting of strip was more obvious. What's more, the effect of restoring years after cutting on the physical properties of soil surface seemed to be superior to chemical properties, but this was not supported by special theory, which was directly reflected from the number of indicators. There was no definitive answer of the recovery period to stand disturbances (Zang, R., and Ding, Y, 2009; Griffiths, P. et. al, 2014), but 9 years should

285  not restore forest soil performance unfortunately.

[revised manuscript text omitted]

435 **Figure 1: Strip plots settings. S1, S2, S3 and S4 show cutting strips with width of 6m, 10m, 14m and 18m respectively. A, B and C show that every strip is divided to 3 subplots with different planting seedlings. Control plot is set near the transformation zone, with the width of 20m.**

[Figure]

**Figure 2. The mean value of physical indicators which have significant difference in various years**

[Figure]

[Figure]

[Figure]

**Figure 3. The mean value of physical indicators which have significant difference in various strips**

[Figure]

[Figure]

[Figure]

440 **Figure 4. The mean value of physical indicators which have significant difference in the interaction of year and strip**

[Figure]

**Figure 5. The mean value of chemical indicators which have significant difference in various years**

[Figure]

**Figure 6. The mean value of chemical indicators which have significant difference in various strips**

[Figure]

445

**Figure 7: Comprehensive soil quality index under cutting strips 3, 6 and 9 years after cutting**